# GWAS reveals determinants of mobilization rate and dynamics of an active endogenous retrovirus of cattle

Lijing Tang [1] ✉, Benjamin Swedlund [1,2], Sébastien Dupont[1], Chad Harland [1,3], Gabriel Costa Monteiro Moreira [1], Keith Durkin [1,4], Maria Artesi[1,4], Eric Mullaart [5], Arnaud Sartelet[1,6], Latifa Karim[1,7], Wouter Coppieters[1,7], Michel Georges [1] ✉ & Carole Charlier [1] ✉

Five to ten percent of mammalian genomes is occupied by multiple clades of endogenous retroviruses (ERVs), that may count thousands of members. New ERV clades arise by retroviral infection of the germline followed by expansion by reinfection and/or retrotransposition. ERV mobilization is a source of deleterious variation, driving the emergence of ERV silencing mechanisms, leaving "DNA fossils". Here we show that the ERVK[2-1-LTR] clade is still active in the bovine and a source of disease-causing alleles. We develop a method to measure the rate of ERVK[2-1-LTR] mobilization, finding an average of 1 per ~150 sperm cells, with >10-fold difference between animals. We perform a genome-wide association study and identify eight loci affecting ERVK[2-1-LTR] mobilization. We provide evidence that polymorphic ERVK[2-1-LTR] elements in four of these loci cause the association. We generate a catalogue of full length ERVK[2-1-LTR] elements, and show that it comprises 15% of *C*-type autonomous elements, and 85% of *D*-type non-autonomous elements lacking functional genes. We show that >25% of the variance of mobilization rate is determined by the number of *C*-type elements, yet that de novo insertions are dominated by *D*-type elements. We propose that *D*-type elements act as parasite-of-parasite gene drives that may contribute to the observed demise of ERV elements.

Half of mammalian genomes is composed of interspersed repeats, including endogenous retroviruses (ERVs) that occupy ~5–10% of genome space[1]. ERVs derive from retroviral infection of the germline enabling vertical viral transmission. Such retroviral endogenization may lead to the progressive expansion of a clade of ERV elements that may count tens to thousands of members, by an ERV-encoded reverse transcriptase (RT)-dependent copy-paste mechanism. At first, this process entails ERV-encoded envelope (ENV)-associated budding of viral

particles followed by germline reinfection. Subsequently, the ERV may sometimes at least in part forgo the extracellular phase yet continue to multiply by more efficient intracellular retro-transposition. In mice, such a transition from IAPE to IAP elements has been shown to result from the combined acquisition of a novel GAG addressing signal and loss of ENV. Expansion of endogenized ERV clades may perdure for hundreds of thousands to millions of years, well after extinction of the initiating exogenous retrovirus. Further expansion of ERV clades is eventually

[1]Unit of Animal Genomics, GIGA & Faculty of Veterinary Medicine, University of Liège, Liège, Belgium. [2]Keck School of Medicine, University of Southern California, Los Angeles, USA. [3]Livestock Improvement Corporation, Hamilton, New Zealand. [4]Laboratory of Human Genetics, GIGA & Faculty of Medicine, University of Liège, Liège, Belgium. [5]CRV, Arnhem, The Netherlands. [6]Comparative Veterinary Medicine, FARAH & Faculty of Veterinary Medicine, University of Liège, Liège, Belgium. [7]Genomics core facility, GIGA, University of Liège, Liège, Belgium. ✉e-mail: lijing.tang@uliege.be; michel.georges@uliege.be; carole.charlier@uliege.be

curtailed by the accumulation of ERV-disrupting mutations and the acquisition - by the host - of specific ERV-silencing mechanisms including targeted DNA and chromatin epigenetic modifications, piRNAs, and so-called restriction factors targeting various steps of the ERV life cycle, some of which are co-opted ERV genes. ERV endogenization has been caught in the act at least six times, including in poultry, sheep/goat, mice, cat and koala, indicating that it is a common phenomenon[2–6].

As for other transposable elements, de novo ERV transposition events in the germline increase the mutational load of populations including by causing disease. Deleterious ERVs are hence subject to purifying selection. Conversely, ERV elements provide a substrate for the emergence of new functionalities such as novel *cis*-acting regulatory elements[4,7] and even new genes. As an example, the *syncytin* genes, which are essential for placentation, derive from ancient ERVs. Also, ERV elements may condition the host's susceptibility to exogenous retroviruses, including by providing protection. Hence, ERVs are important drivers of genomic innovation[2–6].

As ERVs settle, expand and then regress in a species' genome, the rate of ERV mobilization is bound to be an evolving phenotype. Presently, ERVs are largely silent in the human germline[8], while several clades are still active in the mouse in which they account for a sizeable fraction of deleterious mutations[9]. To what extent the ERV mobilization rate varies between individuals within species during this process, and what the underlying determinants of this variation may be, remains largely unexplored. Modifiers of epigenetic modification and expression of specific ERV elements have been mapped in mice and polymorphisms in KRAB zinc-finger proteins nominated as plausible candidates[10], yet whether these also affect transposition rate is not known.

We herein take advantage of unique features of domestic cattle, and of advances enabled by combining next-generation sequencing (NGS) with CRISPR/Cas9, to perform genetic analyses of inter-individual variation in ERV mobilization rate in this species.

## Results

### The insertion of an ERVK[2-1-LTR] element in exon 5 of the APOB gene causes cholesterol deficiency in cattle

A new autosomal recessive defect, referred to as cholesterol deficiency (CD), emerged in Holstein-Friesian dairy cattle around 2015 (OMIA:001965-9913)[11–13]. Autozygosity mapping positioned the corresponding locus on chromosome 11[11]. We and others sequenced the

whole genome of acknowledged carriers and showed that CD is caused by the insertion of an ERV element in exon 5 of the *apolipoprotein B* (*APOB*) gene[12–14]. We PCR-amplified and sequenced the corresponding ERV element and showed that (i) it is ~6.8 Kb long, (ii) belongs to the [2-1-LTR] subgroup of ERVK elements[15,16], and (iii) does not contain full-length open reading frames (ORFs) for either the *GAG*, *PRO*, *POL* or *ENV* genes and is therefore non-autonomous. RNA-Seq analysis of liver of an affected animal showed complete transcriptional shutdown in the ERV's 5'LTR, truncating ~ 97% of the *APOB* ORF (Fig. 1a, b; Supplementary Fig. 1). All carrier animals trace back to *Maughlin Storm*, a sire that was popular in the 1990-ies, suggesting that the birth of the CD mutation might be a recent event and, hence, that ERVK[2-1-LTR] might still be mobile in cattle.

### ERVK[2-1-LTR] elements are active in the bovine germline

We previously generated the Holstein-Friesian *Damona* pedigree to study de novo mutations in the bovine germline. It comprises 743 animals constituting 127 (overlapping) three generation pedigrees including at least two parents (sire and dam), one offspring and ≥5 grand-offspring. The whole genome of parents and offspring was sequenced at average 26-fold depth, while that of the grand-offspring was sequenced at average 10-fold depth (Methods). We developed *LocaTER* (*Localization of Transposable Elements and Repeats*), a bioinformatic pipeline for the detection of transposable element insertions that are not present in the reference genome (Methods). *LocaTER* detected 1222 ERV insertions that are polymorphic in Holstein-Friesian. Repbase[15,16] reports four main groups of ERVs: ERVL-MaLR, ERVL, ERV1 and ERVK, jointly accounting for ~4.7% of genome space. While ERVL (ERVL-MaLR + ERVL) are the most abundant group in the reference genome (55.5% of genome space), followed by ERV1 (29.3%) and ERVK (15.2%), ERVK includes the most abundant amongst polymorphic ERV elements (81.5% of polymorphic elements), followed by ERV1 (15.6%) and ERVL (2.9%). This is compatible (assuming approximately equal element size) with ERVK being younger and still active (Fig. 2a). Repbase reports 33 subgroups of ERVK, of which four are overrepresented amongst polymorphic ERVK elements, including ERVK[2-1-LTR] (20.2% of ERVK space, 26.6% of polymorphic ERVK elements) and BTLTR1B (10.2% of ERVK space, 27.1% of polymorphic ERVK elements) (Supplementary Fig. 2a).

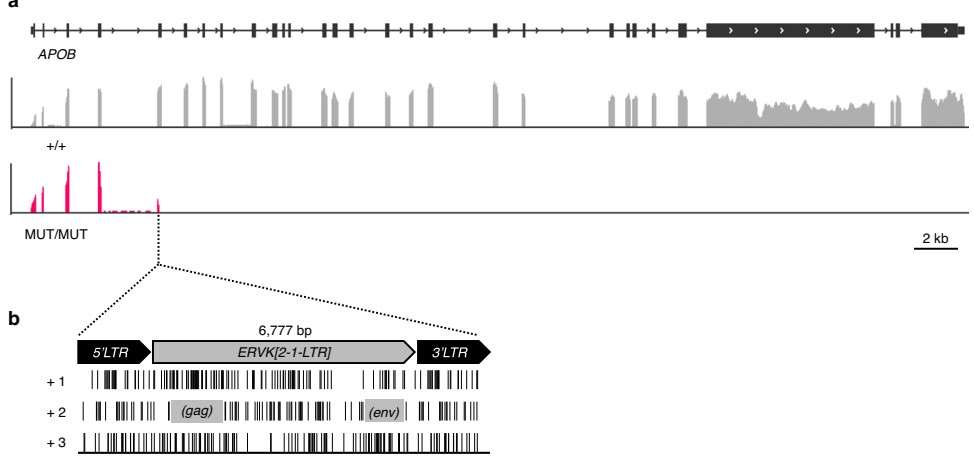

**Fig. 1 | The insertion of an ERVK[2-1-LTR] element in the *APOB* gene causes cholesterol deficiency in Holstein-Friesian cattle. a** Genomic structure of the bovine *APOB* gene (chr11: 77,885,988-77,927,967 bp)(upper lane), and liver cDNA sequence coverage tracks for a homozygous wild-type animal (+/+, grey) and an affected homozygous mutant animal (MUT/MUT, red), highlighting the transcriptional shutdown in exon 5, with partial retention of intron 4 (see also

Supplementary Fig. 1). **b** Schematic representation of the ~ 6.8 Kb ERVK[2-1-LTR] element sense insertion with the translation of its full-length sequence in the three frames (+ 1, + 2, + 3), revealing the absence of intact open reading frames (ORFs) in the proviral sequence flanked by two identical long terminal repeats (LTRs). Stop codons are represented by black vertical bars. Partial *GAG* and *ENV* ORFs are shown as grey rectangles (frame +2).

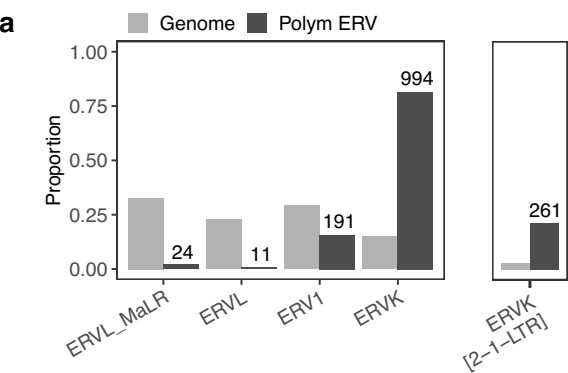

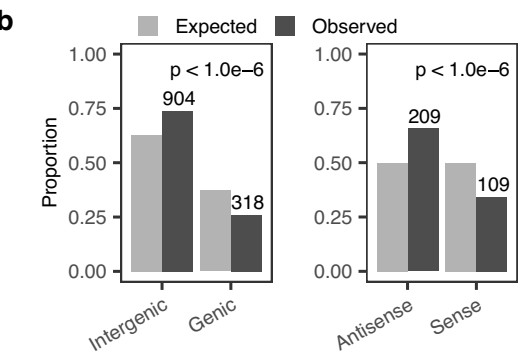

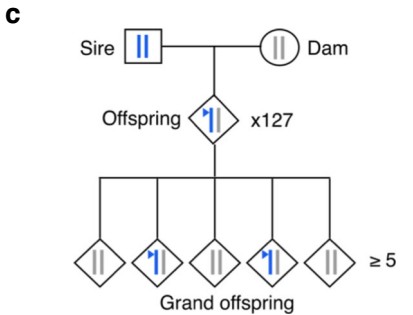

**Fig. 2 | Detection of polymorphic ERVs and de novo ERV mobilization events in the *Damona* pedigree. a** ERVK elements - including the ERVK[2-1-LTR] clade - are overrepresented amongst polymorphic ERVs detected with *LocaTER* when compared to their abundance (relative to other ERV elements) in the bovine genome, supporting their youth. **b** ERVs are overrepresented in intergenic regions, or – when genic – in antisense (as opposed to sense) orientation, when compared to the corresponding genome space, supporting purifying selection. Two-sided *p*-values ($<10^{-6}$) were determined from the outcome of $>10^6$ random samples with probabilities of success corresponding to the genomic expectations (sample() function in R). **c** The *Damona* pedigree comprises 743 whole genome sequenced animals constituting 127 pedigrees with at least three generations: parents, offspring and ($\geq 5$) grand-offspring. De novo mutations (including ERV mobilization events; blue triangle) are detected by their absence in the parents, presence in the offspring, and transmission to some grand-offspring. Linkage with a grand-parental chromosome (sire's blue chromosome in the example) allows assignment of the de novo event to the sire's or the dam's germline. **d** Description of the five de novo ERVK[2-1-LTR] insertions, with, from left to right: (i) chromosomal position (ARS-UCD1.2 genome assembly), (ii) genome compartment, (iii) when genic, orientation with respect to the affected gene, (iv) when genic, gene symbol of the affected gene, (v) parental germline in which the insertion occurred: Dam (blue) or Sire (red) with numbers identifying individuals according to Supplementary Fig. 3 (three de novo insertions occurred in Sire 1; the * identifies the two insertions that were co-transmitted in the same gamete), (vi) Mendelian transmission to "x" (2 or 3) out of five grand-offspring (G-off). Source data are provided as a Source Data file.

Nine hundred and four ERV elements mapped to intergenic regions (expected: 611), 109 within genes in sense orientation (expected: 159), and 209 within genes in antisense orientation (expected: 159) ($p < 10^{-6}$), supporting purifying selection against genic sense insertions (Fig. 2b). These trends did not differ significantly between the ERVK and non ERVK groups ($p_{genic\ vs\ intergenic} = 0.61$; $p_{sense\ vs\ antisense} = 0.42$) (Supplementary Fig. 2b, c). Allelic frequency distribution was mildly shifted towards lower values for genic versus intergenic ($p = 0.06$), but not for genic sense versus antisense insertions ($p = 0.99$) (Supplementary Fig. 2d, e; Supplementary Data 1).

Most interestingly, we detected five ERV insertions that were present in an offspring, transmitted to grand-offspring, but absent in both sire and dam, which we considered to be de novo mobilization events (Fig. 2c). Linkage analysis in the grand-offspring indicated that four of these occurred in the germline of a sire, and one in the germline of a dam. Intriguingly, three of the four male mobilizations occurred in the germline of the same bull, of which two in the same sperm cell. We PCR-amplified and sequenced the five de novo insertions. All five measured ~6.8 Kb and their sequence was very similar to the non-autonomous ERVK[2-1-LTR] insertion in the *APOB* gene (Fig. 2d; Supplementary Fig. 3).

These results confirm that ERVK[2-1-LTR] are presently active in bovine, at a rate of ~ one event per 51 gametes (or ~ one event per 126 gametes when ignoring the exceptional sire), and suggest that the mobilization rate may differ between individuals. Mining of the whole genome sequence of the bull transmitting three ERVK[2-1-LTR] de novo insertions did not reveal striking anomalies in 38 genes that have been connected with control of ERV mobilization[17] (Supplementary Data 2).

## Developing a method to measure ERVK[2-1-LTR] mobilization rate in germline and soma

We adapted Pooled CRISPR Inverse PCR (PCIP)[18] to measure the rate of ERVK[2-1-LTR] mobilization in a sample of genomic DNA. As for PCIP, we used CRISPR/Cas9 to augment the efficacy of inverse PCR, yet used Illumina short read sequencing instead of Oxford Nanopore long read sequencing to increase coverage and accuracy (Fig. 3a). Noteworthy features of the method are: (i) the realization of technical replicates by targeting both 5' and 3' LTR, (ii) the ability to recognize PCR duplicates on the basis of shared shearing sites (SS)(if two sperm cells carry the same de novo ERVK[2-1-LTR] insertion, the probability that the two corresponding DNA molecules are broken at the exact same position is assumed to be very low; shared SS are therefore assumed to correspond to PCR duplicates), and (iii) the ability to express the number of detected de novo mobilization events as a function of the effective number of explored genomes (using inherited ERVK[2-1-LTR] elements as internal controls, see Methods). Of note, ERVK[2-1-LTR] with polymorphisms in the CRISPR target sites may escape detection by PCIP, an

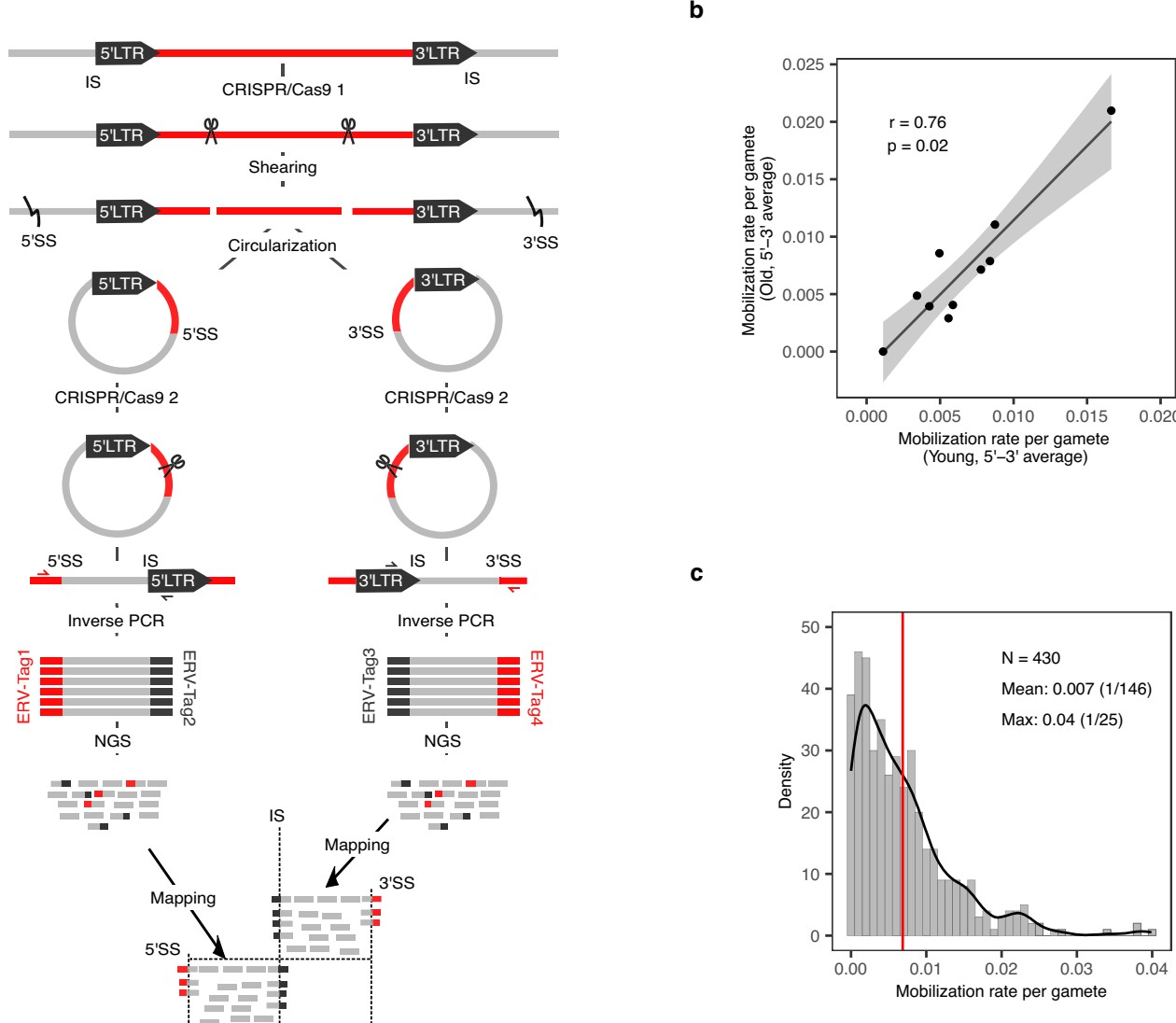

**Fig. 3 | Measuring the de novo rate of ERVK[2-1-LTR] mobilization using an adaptation of Pooled CRISPR inverse PCR sequencing (PCIP-Seq). a** Method: Genomic DNA is attacked with a pair of CRISPRs targeting the ERVK[2-1-LTR] provirus at 429 and 374 bp from the 5' and 3' LTR sequences, respectively. The DNA is then mechanically sheared (generating 5' and 3' shearing sites (SS)), end-repaired, circularized, and attacked separately by a second pair of CRISPRs targeting proviral sequences adjacent to the LTRs to specifically reopen circles encompassing ERVK[2-1-LTR] proviral insertion sites (IS). Fragments encompassing the IS and SS are then amplified by long-range inverse PCR using divergent ERVK[2-1-LTR] targeting primer pairs. The resulting PCR products are subject to NGS and the obtained sequence reads mapped on the reference genome, revealing ERVK[2-1-LTR] element insertion sites (IS), shearing sites (SS) used as molecular identifiers, and short sequence tags that inform about the inserted ERVK[2-1-LTR] element.

PCIP provides information about the genotype of the individual for polymorphic ERVK[2-1-LTR] elements that segregate in the population, the number of genomes that were effectively captured for that individual (based on the number of detected shearing sites for the polymorphic ERVs), and the number of de novo mobilization events detected in the individual's DNA sample. **b** Correlation between estimates of the mobilization rate of ERVK[2-1-LTR] elements in sperm samples collected at > 7 years interval (young vs old) for ten Belgian Blue bulls. Estimates correspond to the average of the 5'LTR and 3'LTR measures. Spearman's correlation was 0.76, with two-sided *p*-value of 0.02. The 95% confidence region for the regression fit was added using the stat_smooth() ggplot function. **c** Frequency distribution of estimated ERVK[2-1-LTR] de novo mobilization rate in sperm of 430 Belgian Blue bulls (red vertical bar = mean). Source data are provided as a Source Data file.

issue which is at least partially mitigated by targeting both 5' and 3' proviral ends. The correlation between 5' and 3' rate estimates was 0.68, and 0.85 between averages of 5' and 3' estimates for biological replicates, testifying for the accuracy of the approach (Supplementary Fig. 4a, b; Supplementary Data 3 & 4).

**The ERVK[2-1-LTR] mobilization rate in the male germline varies between individuals**

We first obtained pairs of semen samples from 10 Belgian Blue (BB) bulls, collected at ≥ 7 years interval. We applied our method to these samples and computed the correlation between ERVK[2-1-LTR]

mobilization rate (5' and 3' average) estimated at young and old age. The correlation, corresponding to the so-called repeatability of the trait (upper bound of the heritability), was 0.76 (Fig. 3b). This implies that the ERVK[2-1-LTR] mobilization rate (i) varies between animals, and (ii) is stable over long periods of time within animal. There was no evidence of an effect of age on ERVK[2-1-LTR] mobilization rate. We detected 260 de novo ERVK[2-1-LTR] mobilization events in these experiments. Their median "dosage" in sperm DNA was 0.00023, or one copy per 4318 sperm cells (range: 0.00018–0.00265). This indicates that the level of mosaicism for these de novo insertions is low, and hence that they mostly occur at late stages of spermatogenesis.

Yet, we observed 38 cases where the same insertion was captured both in the young and old samples. This suggests that the corresponding insertions are present in spermatogonial stem cells that maintain the capacity to give rise to spermatogenic waves during the entire life of the animal (Supplementary Data 5).

We then applied the method to semen samples from 430 Belgian Blue (BB) bulls. We captured a total of three fixed ERVK[2-1-LTR] elements (i.e., detected in all studied animals), 306 polymorphic ones (i.e. detected in some but not all animals), and 3669 de novo ERVK[2-1-LTR] insertions in this experiment (Supplementary Data 4 & 6). De novo ERVK[2-1-LTR] elements tended to preferentially insert near telomeric ends and in GC-rich regions (even after correcting for distance from chromosome end). Insertion sites were characterized by a 6-bp duplication and an 8 bp pseudo-palindromic motif[19] (Supplementary Fig. 5). They resided in genic regions more often than expected by chance ($p = 0.0004$), and – in those – equally often in sense as in antisense orientation ($p = 0.40$). We observed a paucity of de novo insertions on chromosome X given its size, which is likely due to hemizygosity in males (Supplementary Fig. 6). The estimated ERVK[2-1-LTR] mobilization rate averaged one per 146 sperm cells, yet ranging from zero to one in 25 (Fig. 3c, Supplementary Data 4). There was no evidence of an effect of the bulls' inbreeding coefficient on ERVK[2-1-LTR] mobilization rate (Supplementary Fig. 4c).

## GWAS identifies eight loci affecting ERVK[2-1-LTR] mobilization rate

In order to investigate whether the variance in inter-individual transposition rate has a genetic basis, we determined the whole genome sequence (average 40-fold depth) of 40 of the 430 BB bulls and genotyped them at ~14 million variant positions using GATK[20]. The remaining 390 bulls were genotyped using a 50 K medium density Illumina array and genotypes augmented to whole genome by imputation. We kept ~10 million variants with minor allele frequency (MAF) ≥ 0.02, together with the PCIP-deduced genotypes (+/+, +/ERV, ERV/ERV) for 87 polymorphic ERVK[2-1-LTR] elements (MAF ≥ 0.02), for further analyses on the complete dataset (Supplementary Data 7). Of note, 80 of 306 polymorphic ERVK[2-1-LTR] elements were singletons, 163 had a population frequency < 0.05 (rare non-singletons), and 63 a frequency ≥ 0.05 (common). We conducted a GWAS for ERVK[2-1-LTR] mobilization rate using a linear model including a fixed variant dosage effect and a random polygenic effect to correct for stratification. We obtained a major, genome-wide significant signal (-log(p) = 29.7) on chromosome 19 (Fig. 4a). We fitted this effect as covariate in the model and repeated the genome scan. This revealed seven additional genome-wide significant (or near) (-log(p) = 7) signals (Fig. 4b).

Strikingly, a polymorphic ERVK[2-1-LTR] element was either the top variant (chromosome 19) or in high linkage disequilibrium (LD) with it ($r^2 \geq 0.62$; chromosomes 2, 4 and 21) for four of the eight loci (Fig. 4c, Supplementary Fig. 7), a coincidence which is very unlikely to have occurred by chance alone ($p < 10^{-8}$; see Methods).

While it may have been expected that inherited ERVK[2-1-LTR] elements promote ERV mobilization rate, this finding raised the question of what differentiates the ERVK[2-1-LTR] elements in the four associated loci from other polymorphic ERVK[2-1-LTR] elements.

## Genome-wide landscape of polymorphic ERVK[2-1-LTR] elements in Belgian Blue cattle

This question prompted us to extensively characterize the polymorphic ERVK[2-1-LTR] repertoire in BB cattle. As mentioned before, PCIP coamplified a total of 309 endogenized ERVK[2-1-LTR] elements from the semen samples of the 430 BB bulls. We successfully PCR-amplified and completely sequenced 221 of these (Supplementary Data 8). This revealed two subclades, differing by poorly alignable central fragments of 6.2 (C-type) or 2.7 Kb (D-type), and accounting respectively for 15% and 85% of PCIP-detected ERVK[2-1-LTR] elements

(Fig. 5a and Supplementary Fig. 8a). Intact ORFs for *GAG*, *PRO*, *POL* and *ENV* were present for ~50% of C-type members, hence assumed to be autonomous or *C*ompetent, while being absent for all *D*-type members, hence assumed to be non-autonomous or *D*efective (Supplementary Fig. 9).

We detected 216 substitutions (i.e. differences between ERVK[2-1-LTR] elements) within the C-specific fragment (of which 60 singletons), 187 substitutions within the D-specific fragment (45 singletons), and 391 substitutions in the 5' and 3' flanking regions (123 singletons). There were ~6 times more CpG to TpG and CpG to CpA substitutions than expected, reflecting the increased mutability of methylated cytosines. There were more G-to-A substitutions than C-to-T substitutions at non-CpG sites ($p = 0.0007$), supporting strand-specific APOBEC3-dependent G-to-A editing[21] (Fig. 5b, c). The latter was substantiated by the observation of clustered G-to-A singletons for several endogenized ERVK[2-1-LTR] elements (Supplementary Fig. 10a).

We designed probes targeting the insertion sites of 291 polymorphic ERVK[2-1-LTR] elements, added them to Illumina 50 K arrays, and successfully re-genotyped the 430 bulls for 193 loci. Striking discrepancies between the array and PCIP genotypes were observed for at least five loci. Examination of the whole genome sequences (WGS) of the 40 bulls, revealed that this was due to the segregation of a third solo-LTR allele (in addition to "+" and full-length ERV, where "+" corresponds to the ancestral or wild-type allele) for these loci. Further examination of the WGS data with *LocaTER* output indicated that there exist at least 100 additional polymorphic ERVK[2-1-LTR] elements for which the only two segregating alleles are solo-LTR and "+", hence never captured by PCIP. The presumed solo-LTR status was confirmed by PCR for 84 tested elements, indicating that no important ERVK[2-1-LTR] subclade (other than C and D) was missed by PCIP (Supplementary Data 9). The distribution of allele frequencies in the 40 sequenced bulls was slightly shifted towards higher values for solo-LTRs when compared to C- and D-type ($p = 1.4 \times 10^{-4}$ and $2.0 \times 10^{-8}$), supporting their older age as expected (Supplementary Fig. 11).

It is well established that recombination between ERV genomes can occur during reverse transcription of the pseudodiploid genomic RNAs (gRNAs)[22]. We examined the haplotype patterns of the 221 ERVK[2-1-LTR] elements to search for evidence of recombination. When a new variant arises by mutation (f.i. + becomes *M*), it is found in "complete" association ($D' = 1$) with the variants (f.i. $V_1$) constituting the ERV element upon which it occurred: only three of the four possible haplotypes exist across all elements ($V_1$-*M*, $V_1$-+ and +-+). The appearance of the fourth haplotype (+-*M*) requires a recombination between a gRNA with $V_1$-*M* haplotype and one with +-+ haplotype. We first conducted this "four haplotypes" test for all pairs of variants separately for the C and D subclades. There was ample evidence for past recombinations having occurred throughout the ERVK[2-1-LTR] genome both within the C and within the D subclades, indicating that gRNA originating from distinct ERVK[2-1-LTR] elements (albeit from the same subclade) can be co-packaged in the same virus-like particle (VLP) and recombine (Fig. 5d). We then looked for evidence of recombination between C- and D-type ERVK[2-1-LTR] elements. We did this by examining the linkage disequilibrium (LD) between C vs D genotype and all variants in the 5' and 3' flanking segments. Here also, there was ample evidence for past recombinations between C-and D-type gRNA indicating that these can also be co-packaged in VLP (Fig. 5e and Supplementary Fig. 10b). LD with C/D genotype was nearly perfect ($r^2 \sim 1$) for a set of variants immediately flanking the subclade-specific segments, suggesting that recombination between C and D gRNAs might be hampered in close proximity to the point of C-D divergence, leading to the progressive displacement of the C-D boundaries in a manner reminiscent of the "attrition" observed at mammalian pseudoautosomal boundaries[23] (Fig. 5e and Supplementary Fig. 8a).

The consensus GAG, POL (reverse transcriptase domain) and ENV (transmembrane domain) sequences of C-type elements each cluster

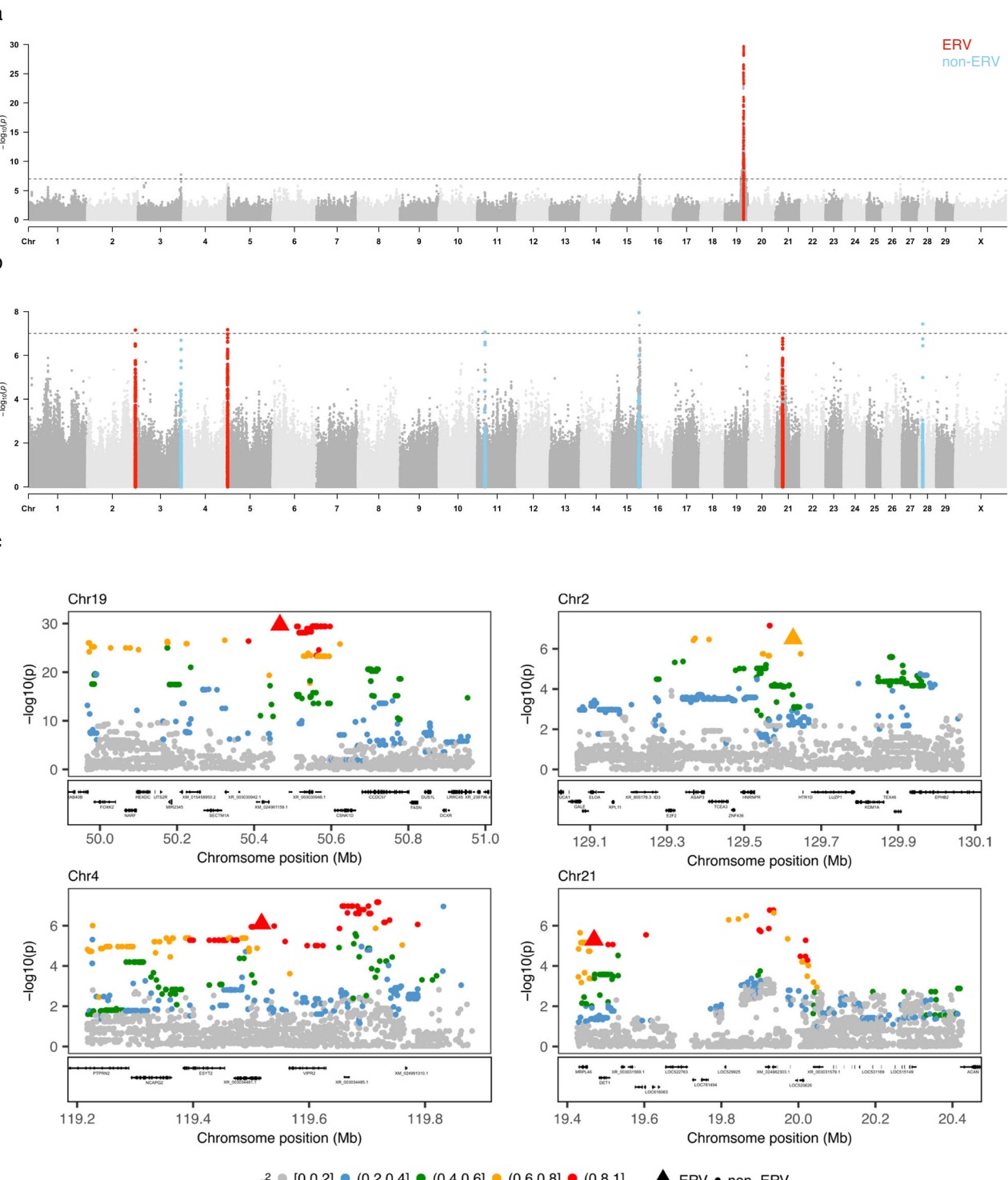

**Fig. 4 | GWAS for the rate of de novo ERVK[2-1-LTR] mobilization in the male germline of cattle. a** GWAS conducted using PCIP-determined mobilization rate in sperm samples of 430 Belgian Blue bulls and genotypes at ~10 million SNPs, revealing a very strong signal on chromosome 19. **b** GWAS conducted using the same population after correcting ERVK[2-1-LTR] mobilization rate for the effect of the chromosome 19 QTL. Seven additional (near) genome-wide significant effects were detected. Loci encompassing an ERVK[2-1-LTR] element are highlighted in red, others in light blue. **c** Zoom into the four loci encompassing an ERVK[2-1-LTR] element, shown as triangles (as opposed to circles for SNPs). Variants are colored according to their LD ($r^2$) with the lead variant. The LD between the ERVK[2-1-LTR] element and the lead SNP was 1.00 (ERV = lead variant) for chromosome 19, 0.62 for chromosome 2, 0.84 for chromosome 4, and 0.94 for chromosome 21. All non-ERVK lead variants were imputed variants with imputation accuracy ≥ 0.95. Two-sided *p*-values were obtained with GEMMA as described in Methods. Experiment-wide 5% significance thresholds (dotted horizontal line in (**a**) account for the realization of 500,000 independent tests (see Methods). Source data are provided as a Source Data file.

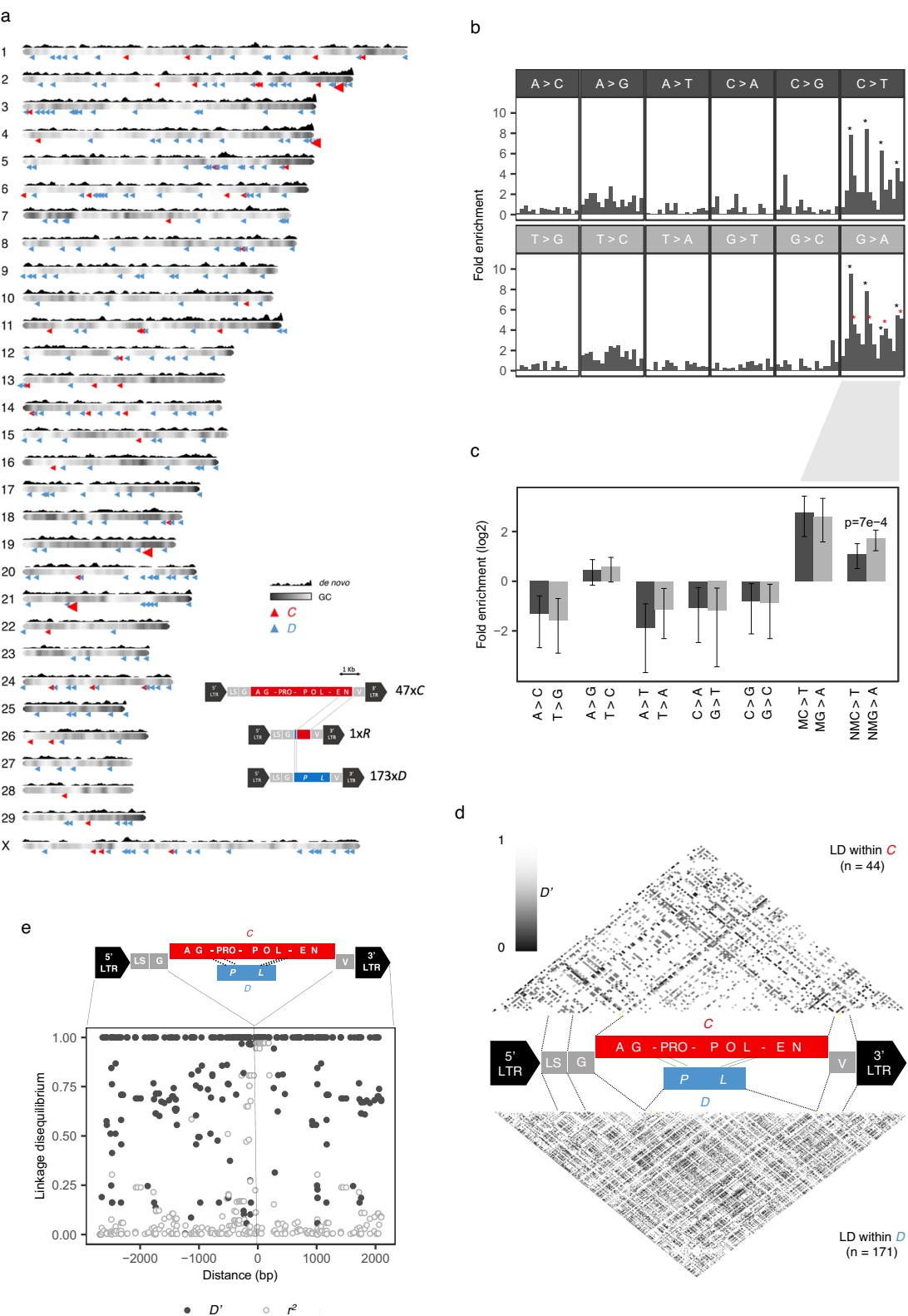

with betaretroviridae (Supplementary Fig. 12a). The ENV protein encompasses a surface unit (SU), proteolytic processing site, and transmembrane unit (TM) with fusion peptide, $CX_7C$ cysteine motif, transmembrane region (TR) and cytoplasmic tail (CT) domains, typical of beta- and lenti-retroviruses[24]. The amino-terminal end of the GAG protein was predicted by Myristoylator[25] to correspond to a myristoylation site (high confidence score of 0.98) (Supplementary

Fig. 12b). Taken together this suggests that ERVK[2-1-LTR] mobilize by inter-cellular reinfection[26]. The consensus primer binding site (PBS), shared by C-type and D-type elements (Supplementary Data 8), presents striking complementarities with the 3' extremities of the two bovine lysine tRNAs (CTT and TTT codons, respectively), supporting the ERVK denomination (Supplementary Fig. 12c). Observed mismatches might reflect the required balance between adequate tRNA

**Fig. 5 | Catalogue of polymorphic ERVK[2-1-LTR] elements segregating in the Belgian Blue Cattle population. a** Resequencing 221 of 309 endogenized ERVK[2-1-LTR] elements, detected by applying PCIP to 430 bulls, revealed a subclade of ~10 Kb *Competent (C)* (15%) and a subclade of ~6.8 Kb *Defective (D)* elements (85%), as well as one *Recombinant (R)* element (inset). Their genome-wide localization is represented by blue *(D)* or red *(C)* small triangles and the four *C* significant GWAS loci by larger red triangles. GC content and density of de novo ERVK[2-1-LTR] insertions along chromosome lengths are shown. **b** 797 single base pair substitutions detected by aligning the sequences of the 221 ERVK[2-1-LTR] elements, sorted by type of substitution and trinucleotide context (Upper panel: ASA, ASC, ASG, AST, CSA, CSC, CSG, CST, …; lower panel: TST, GST, CST, AST, TSG, GSG, CSG, ASG, … where S = Substitution). The Y axis corresponds to the observed fold enrichment accounting for the abundance of the corresponding trinucleotide in the ERVK[2-1-LTR] consensus sequence. The strongest enrichments are observed for NCpG to NTpG and CpGN to CpAN (black asterisks), as expected given the increased mutability of methylated cytosines. The second strongest signal is AGN to AAN (red

asterisks) which is likely an APOBEC3 G-to-A editing signature. **c** Comparison of the enrichment of "mirror" substitutions to test the strand-specificity of the underlying process. Only G-to-A vs C-to-T substitutions at non-CpG sites showed a significant difference in enrichment ($p = 0.0007$, obtained by bootstrapping, two-sided) in support of a role for the strand-specific APOBEC3 dependent G-to-A editing. The error bars mark the 95% confidence interval of the estimates obtained by boot-strapping. **d** Evidence (four gametes rule) of past recombination events between *C*-type (upper triangle) and *D*-type (lower triangle) ERVK[2-1-LTR] elements, respectively. *D'* values ≤ 1 for a pair of variants testify of past recombination events in the corresponding interval. Black or grey points in the upper and lower triangles mark variant pairs with evidence for recombination. **e** Evidence of past recombination events between a *C-* and a *D*-type elements in segments flanking the *C-* and *D*-type specific regions. Solid black dots - D': all values ≤ 1 are a testimony of past recombinations. Empty grey dots - $r^2$: a number of variants in the immediate vicinity of the *C-D* boundaries are in perfect LD ($r^2 \sim 1$) suggesting attrition of the boundary. Source data are provided as a Source Data file.

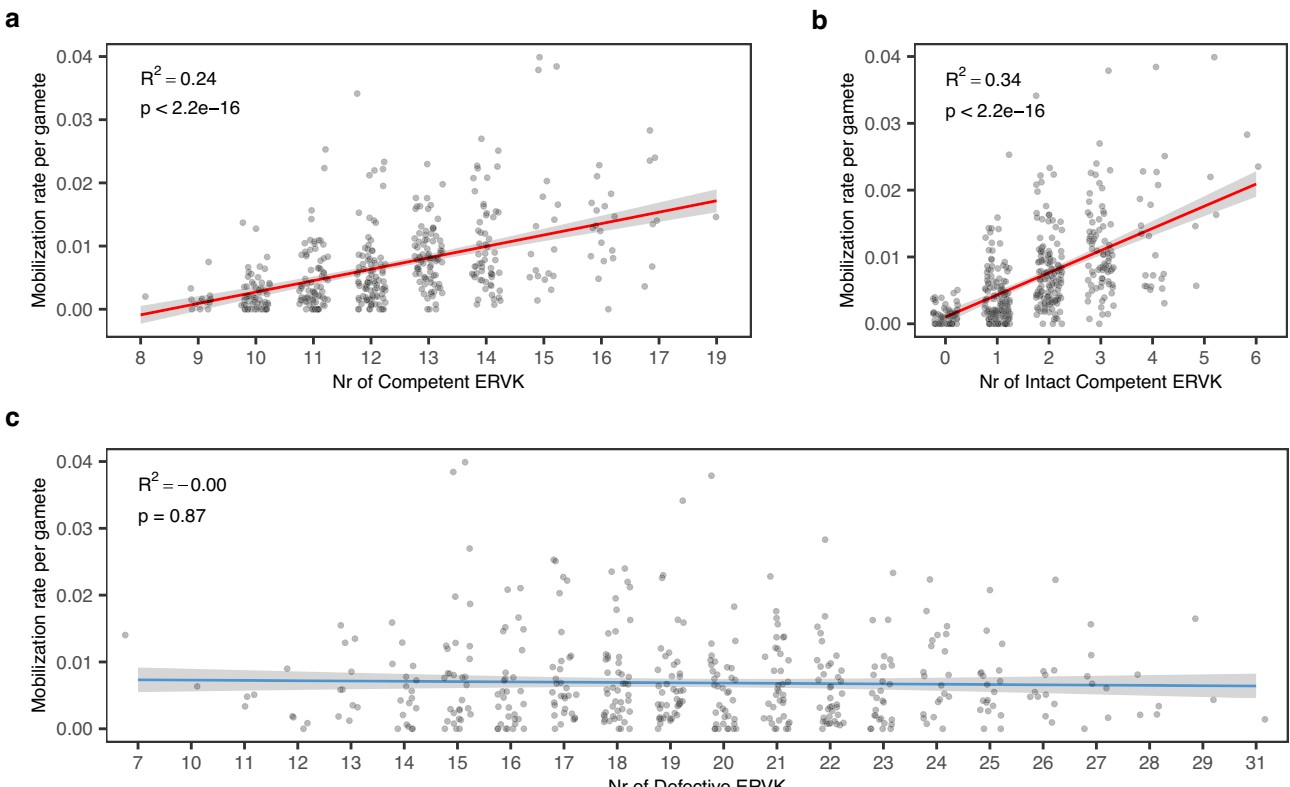

**Fig. 6 | Effect of the number of inherited ERVK[2-1-LTR] elements in the genome on the de novo mobilization rate in the male germline. a** Effect of the total number of ~10 Kb *Competent* elements ($p < 2.2 \times 10^{-16}$). **b** Effect of the number of intact (i.e. without any coding variant in *GAG*, *PRO*, *POL* or *ENV*) ~10 Kb *Competent* elements ($p < 2.2 \times 10^{-16}$). **c** Effect of the total number of ~6.8 Kb *Defective* elements

($p = 0.87$). p: statistical significance (two-sided) of Spearman's correlation between the number of elements and the mobilization rate. $R^2$ = variance explained by the linear regression of number of elements on mobilization rate. The 95% confidence region for the regression fit was added using the stat_smooth() ggplot function. Source data are provided as a Source Data file.

priming for effective reverse transcription yet reduced complementarity to tRNA fragments mediating silencing[27].

## The number of inherited competent ERVK[2-1-LTR] elements explains > 25% of interindividual variation in male mobilization rate

Analysis of the ERVK[2-1-LTR] catalogue showed that the four ERV elements in the GWAS peaks were all of *C*-type ($p = 0.0008$). This suggests that the number of competent ERVK[2-1-LTR] elements in the genome is a major determinant of their mobilization rate. To more accurately evaluate the effect of the number of inherited *C*-type members on ERVK[2-1-LTR] mobilization rate, we compiled this number for the 430 BB bulls considering not only the four loci identified by

GWAS, but all other (rarer and monomorphic) competent elements as well. It ranged from 8 to 19. We tested its effect on ERVK[2-1-LTR] mobilization rate using a linear model. It was highly significant ($p \le 2.2 \times 10^{-16}$) and explained 24% of the variation for this trait (Fig. 6a). In striking contrast, the number of non-competent *D*-type elements, which ranged from 7 to 31, had no effect at all on ERVK[2-1-LTR] mobilization rate ($p = 0.87$; Fig. 6c).

Twenty-five *Competent* ERVK[2-1-LTR] elements harbor coding variants in at least one of the *GAG*, *PRO*, *POL* and/or *ENV* ORFs. This includes a stop codon at amino acid position 13 of the *POL* gene that is carried by the ERVK[2-1-LTR] element coinciding with the GWAS peak on chromosome 4 (Supplementary Data 10). We repeated the analysis of the effect of the number of inherited *C* elements on the ERVK[2-1-

LTR] mobilization rate, considering only "intact" *C* elements (all are polymorphic). The number of intact *C* elements in the genome ranged from 0 to 6. The effect on mobilization rate became even stronger, now explaining 34% of the trait variance (Fig. 6b). To more accurately evaluate the effect of the coding variants on the de novo mobilization rate, we tested the effect of the number of such elements (with ORF-disrupting mutation in *GAG*, *PRO*, *POL* or *ENV*) in the genome (0, 1 or 2), conditional on having 0, 1, 2 or 3 copies of intact *C*ompetent elements (without ORF-disrupting mutation in *GAG*, *PRO*, *POL* and *ENV*). In the absence of any intact *C*ompetent element, increasing the number of copies of ERVK[2-1-LTR] elements carrying a coding variant did not increase the mobilization rate. However, if the genome additionally harbored one or more intact *C*ompetent ERVK[2-1-LTR] elements, the copies with coding variants affected the de novo transposition rate ($p = 1.5 \times 10^{-4}$), pointing towards a possible ($p = 0.11$) epistatic interaction between mutated and intact *C*-type elements (Supplementary Fig. 13).

### Preferential de novo mobilization of defective ERVK[2-1-LTR] elements

The role of competent ERVK[2-1-LTR] elements in driving mobilization rate contrasts with the observation that the ERVK[2-1-LTR] element disrupting the *APOB* gene, and the five de novo mobilized ERVK[2-1-LTR] elements detected in the *Damona* pedigree are all of non-competent *D*-type. To verify whether this was mere coincidence, we selected three BB bulls for which the constitutive *C*- and *D*-type ERVK[2-1-LTR] elements could be best discriminated based on the sequence tags obtained by PCIP (Fig. 3a and Supplementary Data 11, 12). The three bulls harbored 26 (bull 1), 34 (bull 2) and 33 (bull 3) inherited (PCIP-detectable, i.e. ignoring solo-LTRs) ERVK[2-1-LTR] elements in their genome, of which 12, 11 and 13 of *C*-type. PCIP tag information discriminated 13, 17 and 17 haplotypes, of which 2, 2, and 2 included *C* and *D* elements, while all others (11, 15, and 15) were either pure *C* or pure *D* (Fig. 7a–d). We performed 9, 6 and 9 PCIP experiments per bull, revealing respectively 178, 73, and 147 de novo insertions. We estimated the proportional contribution of the different inherited ERVK[2-1-LTR] elements to the de novo insertions by expectation-maximization (EM)(Methods). In all three bulls, *D* elements accounted for a significantly ($p \leq 0.01$) higher proportion of de novo insertions than of inherited *D*-type ERVK[2-1-LTR] elements (Fig. 7e–g). This suggests that *D* elements are able to outcompete *C* elements despite the fact that the latter are driving de novo mobilization supposedly by *trans*-complementation[28].

Strikingly, distinct haplotypes are not contributing equally to de novo events (Fig. 7b–d). For example, the *D*-only A23 and A33 haplotypes are jointly contributing 64.3% of de novo events while they are accounting for only 10.7% of inherited ERVK[2-1-LTR] elements. Also, the mixed A41 haplotype contributes 31.9% of de novo insertions, although it accounts for only 13.9% of inherited ERVK[2-1-LTR] elements. Of note, the A41 haplotype contributes 95.8% of putative *C*-type de novo insertions, while only accounting for 50% of constitutive *C*-type ERVK[2-1-LTR] alleles. Taken together, and even if we conservatively consider that all A41 de novo insertions are of *C*-type, A23 and A33 *D*-type de novo insertions are outcompeting A41 ones 2.6 to 1. It is noteworthy that we also observed five de novo insertions whose PCIP-tags didn't match any of the inherited ERVK[2-1-LTR] elements of the cognate bull, yet could have arisen by recombination.

Amongst the 398 de novo insertions detected in the three bulls, 304 were sampled once, 63 twice, 17 three times, 9 four times, and 5 more than four times (Fig. 8a and Supplementary Data 13). The frequency distribution of resampling rate informs about the developmental window during which de novo ERVK[2-1-LTR] mobilization is most likely to occur (Methods). We compared the real distribution, with the distribution obtained by simulations conducted under various timing scenarios, yet matching the real data for de novo insertion frequency (average number of de novo insertions per sperm cell) and number of explored haploid genomes. The results are compatible with de novo mobilization occurring during a window of ~ 5–9 consecutive cell divisions during the second half (in terms of number of cell division) of spermatogenesis, which may coincide with a period of reduced genome methylation[29] (Fig. 8a, b, Supplementary Fig. 14 and Supplementary Data 13). This is in striking contrast with de novo point mutations of which at least 30% were shown to occur during early cleavage embryonic cell divisions in bulls[30].

### Self-regulation of the number of ERVK[2-1-LTR] elements in the genome?

It is generally assumed that, after initial expansion, families of ERV elements progressively regress as host defense mechanisms emerge. We expected that GWAS might reveal components of these host defense systems, such as clusters of polymorphic piRNA targeting young ERVs. Yet, we could not identify obvious candidate defensive genes in the vicinity of the four associated loci without ERVK[2-1-LTR] elements (Supplementary Fig. 7). Moreover, the minor alleles (and hence more likely derived allele) in these four loci were systematically increasing rather than decreasing mobilization rate. These findings leave open the question of the nature of ERVK[2-1-LTR] neutralizing mechanisms.

The observation that *D*-type ERVK[2-1-LTR] elements outcompete *C*-type ones, suggests an alternative mechanism that may contribute to the demise of ERV families. Indeed, as the number of defective ERV elements takes over the family at the expense of competent copies, which are themselves undergoing the assault of de novo mutations, the family may eventually lose all its competent elements and hence die out. We performed simulations to explore this hypothesis[31]. We assumed (i) that the mobilization rate in the germline of an individual is increasing with the number of competent elements (as observed), (ii) that elements undergo mutations that can only render them defective, but can either increase or decrease their affinity for the mobilization machinery, and (iii) that fitness is decreasing with or independent of the number of ERV elements in the genome (Methods). Under various combinations of parameter values, we systematically observed that defective elements with higher affinity for the mobilization machinery would emerge by mutation, and rapidly displace the competent elements, precluding further expansion of the family. From thereon, the impact on fitness determined the speed at which the family was eventually eliminated from the population (Fig. 9 and Supplementary Fig. 15).

## Discussion

### The polymorphic ERVK[2-1-LTR] clade occupies ~400 loci in the bovine genome

In this work, we have performed an in-depth characterization of a specific clade of bovine ERVs referred to by Repbase[15] as ERVK[2-1-LTR]. By combining whole genome and targeted sequencing, we show that members of this clade occupy ~400 loci in BB cattle, and that a large fraction of those are polymorphic in this population. We show that ERVK[2-1-LTR] elements come in three "morphs": solo-LTR, *C*(ompetent)-type elements and *D*(effective)-type elements. The majority of loci are biallelic, i.e. characterized by the wild-type "+" allele and one "ERV" allele which can be of *C*- (~9% of loci), *D*- (~47%), or solo-LTR-type (~41%). A minority of loci are triallelic (~2%), characterized by the cosegregation of "+" allele, a full-length (*C*- or *D*-type) ERV allele, and the derived solo-LTR allele (Supplementary Fig. 11). Of note, solo-LTRs are expected to be older than their full-length counterparts. Accordingly, their frequency spectrum is shifted upwards. The estimated proportions of *C*-, *D*-, and solo-LTR-type loci therefore change with sample size: as sample size increases, more new *C*- and *D*-type than solo-LTR-type loci are uncovered, hence their proportion is increasing at the expense of the solo-LTR class.

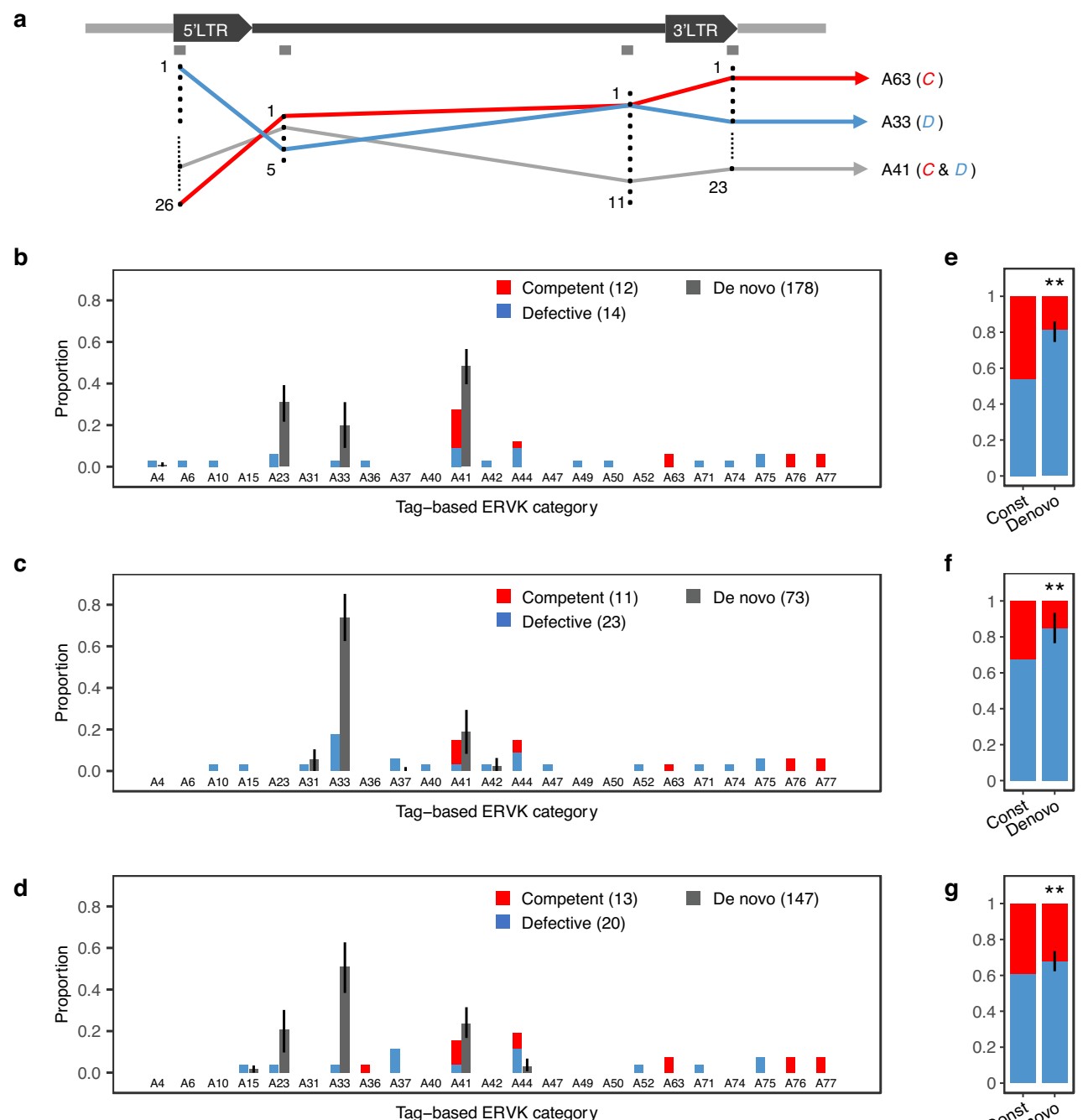

**Fig. 7 | Preferential mobilization of *D*-type ERVK[2-1-LTR] elements. a** Position of - PCIP-tags. Each tag is characterized by multiple (from 5 (tag 2) to 26 (tag 1)) variants across the 309 detected ERVK[2-1-LTR] elements. The 309 elements are each characterized by a combination of variants, i.e. a haplotype. We observed 77 haplotypes (A1 to A77), indicating that different ERVK[2-1-LTR] elements share the same haplotype. For a given animal, some haplotypes correspond to only one ERVK[2-1-LTR] element in its genome (which may be either *C* or *D*), while others may correspond to multiple ERVK[2-1-LTR] elements (which can be all *C*, all *D* or mixed *C* and *D* - such as A41 in the example). Different haplotypes may share the same variant for some tags (A63 and A33 share the same variant for tag 3 in the example). We often have partial PCIP-tag information for de novo insertions (f.i. only the 5′ or 3′ LTR) blurring the assignment of a de novo insertion to a specific haplotype. We used an expectation-maximization (EM) algorithm to probabilistically estimate the proportional contribution of the different haplotypes to the de novo insertions. **b–d** ERVK[2-1-LTR] elements in the genome of three bulls were assigned to 13 (A), 17 (B) and 17 (C) haplotypes, for a total of 23 distinct haplotypes.

The proportion of inherited ERVK[2-1-LTR] elements in each haplotype, as well as the proportion of *C*- (red) and *D*-type (blue) elements in each haplotype, was deduced by combining PCIP and targeted sequencing data. The proportional contribution of each ERVK[2-1-LTR] haplotype to the de novo insertions was estimated by expectation-maximization (EM). As *C*- versus *D*-status cannot be deduced from PCIP-tag information alone, de novo insertions are in dark grey. The 95% confidence interval of the estimates (black lines) was determined by bootstrapping. **e–g** Proportion of *C* and *D* elements amongst endogenous ("Constitutive", red) and de novo inserted ("Denovo", blue) ERVK[2-1-LTR] elements. De novo insertions mapped to mixed (*C* and *D*) haplotypes were distributed amongst *C*- and *D*-types according to the corresponding ratio of inherited elements. The black vertical bars in "Denovo" correspond to the 95% confidence interval determined by bootstrapping. The difference (two-sided) between the C/D ratio for inherited and de novo insertions was significant (** meaning *p* < 0.01). Source data are provided as a Source Data file.

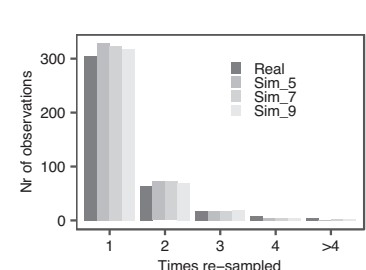
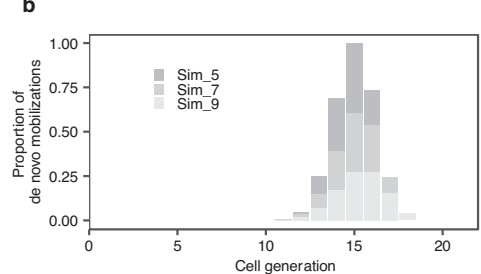

**Fig. 8 | Determining the developmental window during which de novo ERVK mobilization occurs from the frequency distribution of resampling rate.**
**a** Number of de novo insertions captured 1x, 2x, 3x, 4x and >4x for real data (darkest grey) and data simulated using the developmental windows shown in (**b**).
**b** Stacked distributions of de novo mobilization rate at various cell generations of spermatogenesis (1 to 21, i.e. 20 cell divisions generating $2^{20} = 1,048,576$

"spermatogonial stem cells" from one ancestral primordial germ cell) that maximize the correspondence between simulated and real number and resampling rate of captured de novo insertions. Dark grey: 5-generation window; Grey: 7-generation window; Light grey: 9-generation window. The three window-sizes, when centered on cell generation 14, matched the real data (nearly) equally well (Supplementary Data 13). Source data are provided as a Source Data file.

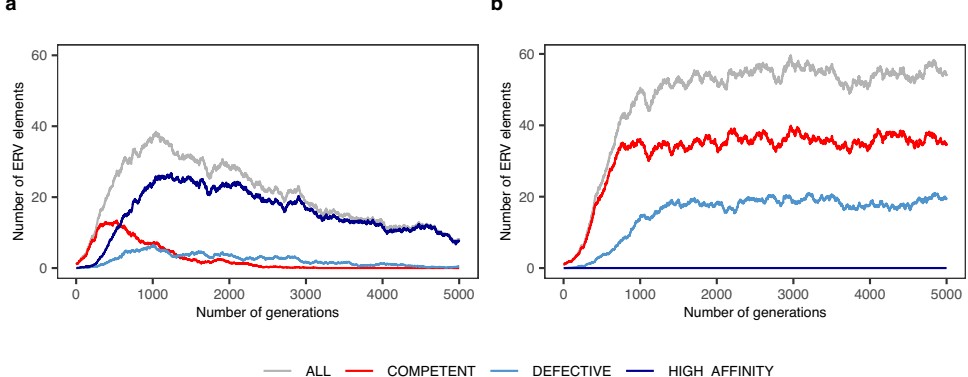

**Fig. 9 | Possible contribution of defective yet high affinity elements to the demise of ERV families. a** Evolution of the average number of ERV elements in the genome (grey: total; red: *C*ompetent; light blue: *D*efective low affinity; dark blue: defective high affinity) of a panmictic population with 1000 individuals, over 5000 generations, with mutation and the possibility to generate *D*efective elements yet with high affinity for the mobilization machinery (provided in *trans* by *C*ompetent elements), and purifying selection. *C*ompetent elements (red) are rapidly overtaken by high-affinity *D*efective ones (dark blue) and die out. The remaining *D*efective

elements (high- and low-affinity, dark and light blue) are progressively eliminated from the genome by purifying selection. **b** Same as in (**a**), yet without the possibility to generate high-affinity *D*efective elements. *C*ompetent elements are maintained in the population at average numbers that are determined by the stringency of the purifying selection. The ratio between *C*ompetent (red) and *D*efective ones (light blue) is determined by the mutation rate. Source data are provided as a Source Data file.

## The ERVK[2-1-LTR] clade derives from ancient endogenization of an unknown exogenous retrovirus

That a high proportion of ERVK[2-1-LTR] loci are polymorphic suggests that their colonization of the bovine genome is a relatively recent event. However, contrary to recent episodes of ERV endogenization reported in poultry, mice, sheep/goat, koala and cat[5], the identity of the exogenous retrovirus at the origin of the bovine ERVK[2-1-LTR] family remains unknown and may be extinct. Sequences with similarities ≥ 95% over ≥ 95% of the full length of the chromosome 19 *C*-type element exist in the reference genomes of other domesticated taurine (*Bos taurus*) and indicine breeds (*Bos indicus*), gaur (*Bos gaurus*), gayal (*Bos frontalis*), domestic yak (*Bos grunniens*), wild yak (*Bos mutus*), bison (*Bison bison*), water buffalo (*Bubalus bubalis*), and African buffalo (*Syncerus caffer*). A marked drop in similarity is observed when querying the genomes of caprinae, including sheep (*Ovis aries*) and goat (*Capra hircus*) (Supplementary Data 14). This suggests that ERVK[2-1-LTR] endogenization may have occurred in an ancestor of Bovinae, i.e. ~15 million years ago. To gain additional insights in the history of the ERVK[2-1-LTR] clade we compared the shared *GAG* and *ENV* sequences between *C*- and *D*-type elements. We restricted the analysis to regions immediately flanking the boundaries of the *C*- and *D*-specific segments and presenting little or no evidence of

recombination (Supplementary Fig. 8). The average pairwise difference between *C*- and *D*-type elements was 38.4 in 250 base-pairs. Assuming a de novo mutation rate of $1 \times 10^{-8}$ base pairs per generation[30], this would correspond to ~15 million generations or ~45 million years. However, this figure doesn't account for the extra mutations introduced during the undetermined number of reputedly low-fidelity reverse transcription steps that separate the different *C*- and *D*-type elements considered in this analysis. It is therefore bound to be overestimated, possibly grossly. We also analyzed the sequence divergence between the 5' and 3' LTRs of all *C*- and *D*-type elements. Upon creation of a new insertion, 5' and 3'LTR have identical sequence. Observed differences therefore reflect the accumulation of de novo mutations and therefore presumably provide information about the time of insertion. The full-length ERVK[2-1-LTR] element with the most divergent 5' and 3'LTRs (98.3% similarity over 1287 base pairs) is a *C*-type element on chromosome 6, present in the bovine reference genome and fixed in the Belgian Blue cattle population (Supplementary Data 8). Assuming a mutation rate of $1 \times 10^{-8}$ base pairs per generation, this level of divergence is expected to accrue over ~ 622,000 generations or ~1.9 million years, hence supporting a more recent time of primordial endogenization of the ERVK[2-1-LTR] clade than the two previous estimates. It should be noted, however, that the eldest

ERVK[2-1-LTR] elements, whose 5' and 3'LTR comparison would best inform about the time of origin of the clade, are more likely to have been reduced to solo-LTRs. This third figure is therefore liable to be an underestimate.

### ERVK[2-1-LTR] elements are still actively mobilizing by reinfection of the bovine germline

Approximately 50% of *C*-type elements have intact ORF for *GAG*, *PRO*, *POL* and *ENV*, suggesting that ERVK[2-1-LTR] might still be active in the bovine germline. This prediction was further supported by the discovery that the insertion of an ERVK[2-1-LTR] element in the *APOB* gene underpins cholesterol deficiency in HF cattle. In this work, we unambiguously demonstrate that ERVK[2-1-LTR] are indeed still active in the bovine male and female germline, first by identifying five de novo mobilization events in whole genome sequenced three-generation pedigrees, and secondly by capturing thousands of de novo insertions in sperm cells by means of PCIP[18]. By analyzing the degree of mosaicism of captured de novo events, we show that ERVK[2-1-LTR] mobilization occurs in late spermatogenesis, yet can affect stem cells that persist throughout the entire life of the animal. The timing of mobilization could coincide with a phase of spermatogenesis during which genome methylation is at its lowest point[17,29]. The observation of an intact *ENV* ORF in a large proportion of *C*-type elements, combined with in silico prediction that the matrix domain (MA) of the GAG protein of *C*-type elements is a target for N-terminal myristoylation (Supplementary Fig. 12), strongly suggests that ERVK[2-1-LTR] elements still multiply by within-host, intercellular reinfection rather than by the supposedly more effective intracellular retrotransposition route. Which cells produce the viral-like particles (somatic or germline cells), which membrane receptor viral-like particles recognize in the recipient cells, and which reinfection path is used (free particles, virological synapses or microtubule-mediated transport) remains unknown.

### The rate of ERVK[2-1-LTR] mobilization is determined by the number of inherited C-type elements

On average, ERVK[2-1-LTR] elements mobilize in the male germline at a rate of ~ one in 150 sperm cells. Yet, this rate varies at least ten-fold between individuals, while remaining remarkably constant over time for a given animal (76% repeatability). We show in this work that the individual ERVK[2-1-LTR] mobilization rate of bulls is determined, to a large extent (r² ≈ 0.25), by the number of inherited *C*-type elements. Intact *C*-type elements (i.e. with uninterrupted ORF for all four *GAG*, *PRO*, *POL* and *ENV* genes) have the most consistent effect on mobilization rate (Fig. 6b). However, even intact *C* elements differ in their effect. For example, an extra copy of the chr19:50466809 ERVK[2-1-LTR] element increases the mobilization rate by 5 events per 1000 sperm cells, while the effects of the chr2:129626969 and chr21:19469667 elements are approximately half or 2.5 events per 1000 sperm cells (Supplementary Fig. 7b). Non-intact *C*-type elements appear to have genuine, albeit more modest effects on mobilization rate, provided that there is at least one intact *C*-type element in the genome (Supplementary Fig. 13). Thus, the effect of the number of non-intact *C*-type elements on mobilization rate depends on the genotype of the animal for intact *C*-type elements, i.e. there is an epistatic interaction between intact and non-intact *C* elements with regards to their effect on mobilization rate. This probably indicates that the concentrations of at least some of the four gene products remain a limiting factor in the mobilization process.

### De novo mobilizations are dominated by trans-complemented D-type insertions

We further show that – despite the driving role of *C*-type ERVK[2-1-LTR] elements – de novo mobilization events are dominated by the insertions of specific *D* elements (Fig. 7). We assume that this reflects (i) the occurrence of *trans*-complementation between *C*- and *D*-type elements, and (ii) a higher affinity of specific *D*- over *C*-type elements for the mobilization machinery. We don't know at this point what the molecular bases of the *trans*-complementation and differential affinity may be. However, the evidence for pervasive recombination between ERVK[2-1-LTR] elements, including between *C*- and *D*-type elements, suggests that *C*- and *D*-type gRNAs can pair in the pseudodiploid virus-like-particles and generate recombinant extrachromosomal DNA (ecDNA) molecules[22]. That *D*-type insertions are able to outnumber *C*-type insertions could either be due to the fact that *D*-type gRNAs are more abundant in cells than their *C*-type counterparts (f.i. because some of them are transcribed at a higher rate or are more stable), or because they are more effective in utilizing the *C*-type provided machinery to generate ecDNA[32]. Another intriguing possibility would be that "heterozygous" virus-like particles harboring a *C*- and a *D*-type gRNA would preferentially produce *D*-type double-stranded extracellular DNA. *Trans*-complementation between mobile elements has been extensively documented before[28]. Examples of complementation in *trans* between morphs of the same ERV clade include ETn by MusD[33,34] and IAP IΔ1 by IAP[35,36] in mice, RecKoRV by KoRVA in koala[37,38], and possibly Type 1 by Type 2 HERV-K in human[39]. Strikingly, the bovine ERVK[2-1-LTR] *C/D* pair, murine MusD/ETn pair, and koala KoRVA/RecKoRV pair share a pattern of swapping of a central segment of the competent ERV element with an old piece of retroelement, yet conservation of flanking sequences encompassing portions of the *GAG* and *ENV* (ERVK[2-1-LTR] and KoRVA) or *GAG* and *POL* (MusD) genes (in addition to the LTRs), which may be suggestive of *cis*-effects on the effectiveness of *trans*-complementation as reported for IAP IΔ1[35].

### D-type elements may act as parasite-of-parasite gene drives

We expected that GWAS for ERVK[2-1-LTR] mobilization rate would reveal emerging components of the host silencing machinery. Yet, there was no obvious evidence for genes participating in such mechanisms in any of the eight detected association peaks. In fact, the minor allele for all top variants (and hence more likely derived allele) was always increasing (rather than decreasing) mobilization rate. This probably indicates the ERVK[2-1-LTR] endogenization is too recent, and that specific silencing mechanisms are yet to evolve in the bovine population. However, our results reveal another mechanism that may precipitate the demise of ERV families. Indeed, *D*-type elements may act as parasite-of-parasite gene drives that may cause the spontaneous implosion of the ERVK[2-1-LTR] clade.

### Possible phenotypic effects of ERVK[2-1-LTR] activity

We show that ERVK[2-1-LTR] mobilization generates deleterious mutations underpinning genetic defects in the cattle population. As shown by the outcome of knocking down host defense mechanisms against transposable elements[17], excessive ERV mobilization rates may compromise fertility, which is an important concern in livestock breeding. To what extent differences in ERVK[2-1-LTR] mobilization rate correlate with fertility in cattle can now be addressed. Also, whether ERVK[2-1-LTR] mobilization affects the transcriptome and contributes to beneficial variation that can be exploited in breeding programs is another interesting question to pursue.

## Methods
### Ethics approval

We used biological materials provided by a breeding program, and the biological materials were collected by veterinarians as a part of routine animal breeding activities, not for experiments. All sperm straws were provided by commercial breeding companies. Hence, our work does not involve animal experiments.

## The Damona pedigree for the detection of de novo ERVK mobilizations

The *Damona* pedigree comprises 743 Dutch Holstein-Friesian cattle assigned to 127 three generation pedigrees including a sire, a dam, an offspring, an average of 8 sibs of the offspring (range: 0–17), and an average of 5 grand-offspring (range:1–11). Moreover, 1.4 grandparents per pedigree are available on average as well (5 pedigrees with 4 grandparents, 16 with 3, 38 with 2, 34 with 1, 34 with 0). The 127 pedigrees overlap: for instance, an animal can be offspring in one pedigree and parent in another. All animals from the *Damona* pedigree were whole genome sequenced (females: blood DNA; males: 24% blood, 76% sperm DNA). The average sequence depth is 26x for sire-dam-offspring trios, 17x for sibs, 10x for grand-offspring and 27.3x for grand-parents.

## LocaTER

The *LocaTER* pipeline scans individual, whole genome sequences (short reads) for insertions of queried interspersed repeats (including ERV elements) that are present in the newly sequenced genome but not in the reference genome to which the reads are mapped (Supplementary Fig. 16). It searches for two features in the sequence data that are characteristic of such insertions. The first is an above normal concentration of discordant paired reads (i.e., the paired ends map to different genome locations) that assort in two adjacent sets. Set A comprises paired reads mapping respectively to the sense strand of the candidate location and to one end of the queried type of repeat (f.i. ERV elements), followed (in the 5′ to 3′ direction on the reference genome) by set B, comprising paired reads mapping respectively to the anti-sense strand of the candidate location and to the other end of the queried repeat. The second characteristic is an above normal concentration of soft- and hard-clipped reads, with (nearly) identical clip position located between sets A and B, and mapping the actual insertion site of the repeat. *LocaTER* also attempts to determine the genotype of the individual for the corresponding insertion, i.e. heterozygous or homozygous. Predictions of *LocaTER* were manually checked (using IGV) for a third characteristic of ERV insertions, namely a short, local target site duplication. A more detailed description of the functioning of *LocaTER* is provided in Supplementary Methods. The *LocaTER* pipeline is available from https://github.com/Lijingtangbo/TE-rate_manuscript.

## Adapting PCIP to quantitatively estimate the rate of TE mobilization

**Molecular biology.** The modified PCIP reaction comprises six steps (Fig. 3a): (i) using 500 ng of genomic DNA as starting material, fragments containing ERVK[2-1-LTR] sequences were cleaved using a pair of single guide RNAs (Integrated DNA Technologies) targeting sequences at 429 and 374 bp from the 5′LTR and 3′ LTR, respectively (Supplementary Data 15), and *S.pyogenes* Cas9 (New England Biolabs, M0386S). Of note, these sequences are the same for the *C*- and *D*-type elements. (ii) The digested DNA was further mechanically sheared to ~3 Kb using a Megaruptor-1 (Diagenode), end-repaired using NEBNext EndRepair Module (New England Biolabs, E6050L), and purified with Agencourt AMPure XP beads (Beckman Coulter, A63881). (iii) The resulting DNA fragments were circularized using T4 DNA Ligase (New England Biolabs, M0202L), residual linear fragments eliminated with Plasmid-Safe-ATP-Dependent DNase (Epicentre, E3110K), purified, and reaction products split in two aliquots. (iv) Circular molecules encompassing ERVK[2-1-LTR] sequences were reopened with *S. pyogenes* Cas9 using distinct single guide RNAs (Integrated DNA Technologies) for the two aliquots (Supplementary Data 15), and purified. (v) ERVK[2-1-LTR] encompassing linear fragments were inverse PCR amplified using aliquot-specific primer pairs (Supplementary Data 15) with LongAmp Taq DNA Polymerase (New England Biolabs, M0533S), and purified. (vi) The purified amplicons were mechanically sheared to

~350 bp using a Bioruptor-pico (Diagenode), sequencing libraries generated using the NEBNext Ultra II DNA Library Prep Kit for Illumina (New England Biolabs, E7645L), and indexed libraries pooled and sequenced on a Novaseq 6000 sequencer (Illumina) targeting ~8 million 150 bp paired-end reads per library.

**Data processing.** The ensuing sequence reads were demultiplexed, quality assessed using fastQC (https://github.com/s-andrews/FastQC), adapter sequences trimmed using Cutadapt (https://github.com/marcelm/cutadapt), and trimmed reads mapped to the bovine reference genome using BWA-MEM40 and converted to BAM format using SAMtools41. Using a custom-made python script, we first identified clipped reads using CIGAR information. We selected clipped reads with mapping quality ≥40 and a minimum of 10 clipped bases. We then mapped the clipped reads to the segments of the ERVK[2-1-LTR] genome corresponding to the ERV-Tags in Fig. 3 (1 and 2 for the 5′LTR libraries and 3 and 4 for the 3′LTR libraries). We demanded an alignment score ≥0.6 to declare a hit, and labelled the read as either an insertion site (IS) or a shearing site (SS) read. When possible, we extended the alignment with ERVK[2-1-LTR] into non-clipped bases to refine the positions of the SS and IS. We then merged SS and IS site with same "breakpoint", thereby identifying candidate SS and IS supported by multiple concordant reads. We then paired IS with their cognate SS. The pairing was based on orientation (f.i. a 5′ SS should be located upstream of the 5′ IS for an ERV element in "sense" orientation, and downstream of the 5′ IS for an ERV in "antisense" orientation; Fig. 3) and distance (the maximum distance between IS and SS was set at 5 Kb). One IS can be paired with one SS (typical for non-mosaic de novo insertions) or with multiple SS (typical for mosaic de novo insertions or inherited ERVK[2-1-LTR] elements). Additional filters were then applied to select candidate de novo insertions for manual inspection, including a distance of more than 3 Kb from constitutive ERVK[2-1-LTR] elements, and only observed in one animal.

**Data normalization.** The above-mentioned procedure yields, for each individual $s$ of $T$, the number of shearing sites for each $i$ of 123 (5′LTR) or 189 (3′LTR) selected "constitutive" ERVK[2-1-LTR] elements ($SSC_{is}$), as well as (at least one) shearing site for $N_s$ de novo ERVK[2-1-LTR] insertions. The $SSC_{is}$'s were modeled (with the R lm() function) as $SSC_{is} = ERV_i + ID_s + \varepsilon_{is}$, where $ERV_i$ is the effect of locus $i$ (not all ERVK[2-1-LTR] elements undergo PCIP as effectively), $ID_s$ is the effect of individual $s$ (we don't engage exactly the same amount of DNA and of same quality in the PCIP reaction for all samples), and $\varepsilon_{is}$ is the error term. We determine the values of $ERV_i$ and $ID_s$ that minimize the error sum of squares. Twice the ensuing $ID_s$ (corresponding to the number of effectively captured haploid genomes for individual $s$) is used as normalization factor to estimate the individual-specific ERVK[2-1-LTR] mobilization rate as $TR_s = N_s / 2ID_s$.

**Software.** Analysis of the sequence reads was conducted using a mixed Python/R pipeline that uses raw sequence data as input and detects both constitutive and de novo ERV integration sites automatically. The pipeline consists of four modules: the mapping module, junction reads annotation module, clustering module and normalization module. It is available from https://github.com/Lijingtangbo/TE-rate_manuscript.

## GWAS for ERVK[2-1-LTR] mobilization rate

**SNP/Indel genotyping and imputation.** Genomic DNA was extracted from sperm using standard procedures. The genome of 40 of 430 Belgian Blue bulls was whole genome-sequenced using Illumina S4 chemistry and a Novaseq 6000 instrument at average sequence depth of 40, and SNPs/Indels detected and genotyped using GATK[20]. The remaining 390 animals were genotyped using Illumina's medium density (~55 K variants) SNP genotyping arrays. For the latter, genotype

information was augmented by imputation in two steps (first to high density using 890 Belgian Blue animals genotyped with Illumina's high density (~770,000 variants) as reference, and then to whole genome using sequenced Belgian Blue animals as reference) using Shapit4[40] for phasing, followed by Minimac4[41] for actual imputation. We kept 10,875,490 variants with minor allele frequency over 2% and imputation accuracy over 90% for GWAS. Four hundred fifteen of the 430 utilized Belgian Blue animals were homozygous for the nt821(del11) mutation in the myostatin (*MSTN*) (gene causing double-muscling in homozygotes[42]), eight heterozygous, and seven homozygous wild-type. *MSTN* genotype had no effect on ERVK[2-1-LTR] mobilization rate.

**Genotyping at 309 constitutive ERVK[2-1-LTR] loci.** We applied the modified PCIP procedure to sperm DNA of all 430 Belgian Blue bulls. Three hundred and nine constitutive ERVK[2-1-LTR] loci were identified as loci marked by high numbers of clustered shearing sites (SS in Fig. 3a) relative to the number of explored haploid genomes (see description of PCIP method above) in at least some animals. Animals were genotyped for the corresponding ERVK[2-1-LTR] loci based on the within-locus (to account for the difference in PCIP efficiency for different ERVK[2-1-LTR]) distribution of the ratio between the number of observed SS and number of explored haploid genomes.

**GWAS.** We conducted GWAS with GEMMA[43] using $TR_S$ as phenotype. The model included a fixed regression on variant dosage (additive effect), as well as a random polygenic effect. Estimating the polygenic effects requires the additive genetic relationship matrix, which was computed from marker data using option "-gk1" in GEMMA. In a second round of GWAS, dosage of the chromosome 19 ERVK[2-1-LTR] locus (i.e. number of alleles with the ERVK[2-1-LTR] insertion) was added as an extra covariate in the model. Linkage disequilibrium in the Belgian Blue population results in the fact that the ~11 million variants behave as ~500,000 independent tests[44], yielding a genome-wide significance threshold of $0.05/500,000 = 10^{-7}$.

**Testing the effect of inbreeding on mobilization rate.** The proportion of autosomal SNPs (MAF $\geq 0.1$) with homozygous genotype was used as proxy for a bull's coefficient of inbreeding. Its effect on mobilization rate was estimated by linear regression using the lm() R function.

**Probability of coincidence of polymorphic ERVK[2-1-LTR] elements and association peaks**
For four of the eight association peaks identified by GWAS, an ERVK[2-1-LTR] element was either the lead variant (1x), or in high LD ($r^2 \geq 0.62$) with the lead variant (3x). The eight corresponding lead variants were characterized by a minor allele frequency (*MAF*) in the 430 bulls which we denote $f_i$. To estimate the probability to observe this level of coincidence or more, fortuitously, we determined, for each lead variant $i$, the proportion of variants (across the entire genome) with $f_i - 0.025 < MAF < f_i + 0.025$ (0.05 bin) that would be in high LD ($r^2 \geq 0.62$) with anyone of the 87 polymorphic ERVK[2-1-LTR] elements with $MAF \geq 0.02$: $p_i$. We then sampled one object in each of eight "urns" (with respective probability of success = $p_i$), repeated this process $10^8$ times, and counted how often we obtained 4 or more "successes". The probability to obtain 1, 2, 3 and > 3 successes was 0.009, $3.9 \times 10^{-5}$, $7.0 \times 10^{-8}$, and zero.

**Establishing a catalogue of ERVK[2-1-LTR] elements in BB cattle Reference sequences for *C*- and *D*-type ERVK[2-1-LTR].** We used ERVK[2-1-LTR] present in the bovine reference genome (ARS-UCD1.2 genome assembly) as reference (*C*-type: chr18-59909149-59919759; *D*-type: chr14-70257004-70263970). We identified the open reading frames (ORFs) corresponding to the *GAG, PRO, POL* and *ENV* genes,

bounded on the amino-terminal end by the first ATG for *GAG, POL, ENV* genes, and by the first non-stop codon in the *PRO* ORF (Supplementary Fig. 9). The corresponding protein sequences were blasted against 'viruses' (taxid: 10239) non-redundant protein sequences to annotate protein domains.

**Determining the full-length sequence of 221 constitutive ERVK[2-1-LTR] elements.** We aimed at amplifying the full-length of all constitutive ERVK[2-1-LTR] detected by PCIP (see above). This was done by amplifying each ERVK[2-1-LTR] element as two overlapping fragments, jointly spanning its full length. Each primer pair (for the 5' half and 3' half of the element, respectively) comprised one primer targeting flanking sequences, and one primer in the body of the ERVK[2-1-LTR] element. As we did not know in advance whether a targeted ERVK[2-1-LTR] element was of *C*- or *D*-type, we tested each flanking primer with *C*-type and *D*-type specific "ERVK body" primers. We performed long-range PCR using the LongAmp Hot Start Taq 2× Master Mix (New England Biolabs, M0533S) and primers listed in Supplementary Data 15. We successfully amplified both left and right halves for 221 ERVK[2-1-LTR] elements, and one half (either left or right) for an additional 74 for a total of 295. This indicates that there are no other important ERVK[2-1-LTR] elements other than the ~6.8 Kb *D*-type and the ~10 Kb *C*-type elements described in this work. The left and right amplicons of the 221 ERVK[2-1-LTR] elements were mechanically sheared to ~500 bp using a Bioruptor-pico (Diagenode), sequencing libraries generated using the NEBNext Ultra II DNA Library Prep Kit for Illumina (New England Biolabs, E7645L) with five cycles of PCR to amplify the adapter-ligated fragments. The quality of the libraries was checked using QIAxcel Advanced System (QIAGEN) and a qPCR. Indexed libraries were pooled and sequenced on a Miseq sequencer (Illumina) targeting 200× coverage. Resulting reads were aligned to cognate (i.e. *C*- or *D*- type, 5' half or 3' half reference) ERVK[2-1-LTR] references using BWA-MEM[42]. We further checked, for each library, whether the corresponding flanking sequences were mapping to the expected genomic coordinates. Variant sites with respect to the reference sequence were annotated with BCFtools[45] and manually curated using IGV[46].

**Identifying ERVK[2-1-LTR] loci with solo-LTR alleles.** We designed probes specific for the + and ERV allele, respectively, for 291 polymorphic ERVK[2-1-LTR] elements, added them to a medium density (MD) SNP genotyping arrays (Illumina), and re-genotyped the 430 bulls. This yielded usable genotypes for 193 ERVK[2-1-LTR] elements. For at least five loci, bulls were genotyped as ERV/+ with the array, while appearing +/+ by PCIP. This suggested the co-occurrence of a solo-LTR allele for these loci, and this hypothesis was confirmed by PCR amplification of the predicted solo-LTR in at least one bull for each one of the five loci. Confronting the results of PCIP-based genotyping of the 40 whole genome sequenced bulls with the results of the analysis of their genome with *LocaTER* revealed 100 ERVK[2-1-LTR] insertions detected by *LocaTER* but never by PCIP. We hypothesized that only solo-LTR alleles would still be segregating in the Belgian Blue population at these loci (i.e. the full-length ERV allele would have been lost). We confirmed the solo-LTR status by PCR amplification and sequencing in at least one bull for 84 of these loci.

**Estimating the proportional contribution of PCIP-tag-defined haplotypes of constitutive ERVK[2-1-LTR] elements to de novo insertions**
PCIP yields two sequence tags of the captured ERVK[2-1-LTR] element on the 5'-side, and two on the 3'-side (Fig. 3a). Polymorphisms in these tags (amongst the 298 endogenized ERVK[2-1-LTR] elements) allow to distinguish 28 5'-side combinations and 39 3'-side combinations, which assort into 77 5'-3' combinations or haplotypes, referred to as A1, A2, … A77 in Fig. 7. The number of endogenized ERVK[2-1-LTR] elements

sharing the same haplotype ranges from 1 to 76. PCIP typically generates 5′-side or 3′-side tag information for de novo insertions, exceptionally (~4%) both (5′-3′). Also, PCIP may only provide information about one (of the two) 5′- or 3′-side tags. Thus, the assignment of a de novo ERVK[2-1-LTR] insertion to a specific existing haplotype is often ambiguous: multiple haplotypes remain possible given the available tag information. To nevertheless be able to accurately estimate the contribution of each of the haplotypes present in the genome of a bull (i.e. the A41, A44, A37's, etc. in Fig. 7) to the captured de novo insertions, we used an estimation-maximization (EM) approach. We started assuming uniform contributions of each endogenized haplotype. As an example, the 15 constitutive ERVK[2-1-LTR] elements inherited by bull A (Fig. 7b) fall into 13 haplotypes (A44, A41, A37, A63, A76, A77, A75, A36, A15, A23, A33, A52, A71). The 13 starting contributions were therefore set at 1/13 ( = 0.077). If a de novo insertion "Z" is compatible with, say haplotype A41 and A44, it is assigned for half to A41 (coefficient 0.5 = 0.077/(0.077 + 0.077)) and for half to A44 (coefficient 0.5). This process is repeated for all de novo insertions, and the contribution of each haplotype computed from the sum of coefficients across all de novo insertions (divided by the number of insertions). After this first iteration, the contributions will have shifted from uniformity. For instance, to $p_{A41}$ for A41 and $p_{A44}$ for A44. The coefficients, for insertion "Z", now become $p_{A41}/(p_{A41} + p_{A44})$ for A44 and $p_{A44}/(p_{A41} + p_{A44})$ for A41. The same process is repeated over all insertions. New contributions are then computed from the sum of updated coefficients across de novo insertions. The process is repeated until convergence (~ 20 iterations).

### Simulating frequency distributions of resampling rate (1,2, 3,… times) of de novo insertions matching experimental data

When, during the development of the germline, the ERVK[2-1-LTR] elements are mobilized is an important question. The degree of mosaicism of de novo ERVK[2-1-LTR] insertions provides information about the timing of the mobilization event. If a de novo insertion has a "dosage" of 50% (i.e. detected once every two studied haploid genomes), it supposedly has occurred at the embryonic one cell stage, if 25% it supposedly has occurred at the two-cell stage, etc. If the dosage is ~ 1/1000 and assuming that there are ~ 1 million spermatogonial stem cells[47], it suggests that it has occurred ten cell divisions before the spermatogonial stem cell stage, yielding a "clone" of ~ 1000 ($2^{10} = 1024$) spermatogonial stem cells sharing the same insertion. If the dosage is ~ 1/1,000,000, it suggests that the mobilization event occurred during the very last cell division(s). A problem is that we only explored between 5,000 and 10,000 haploid genomes for the three bulls for which multiple (6 to 9) PCIP experiments were conducted. Thus, we cannot accurately measure dosages below ~ 1/5,000. Any de novo insertion that is captured once will have an estimated dosage of $1/g$, where $g$ is the number of effectively studied haploid genomes (see above). The actual dosage may be much lower; it could just be that this insertion was "by chance" one of those captured out of a very large pool of very rare de novo insertions. Of note, contrary to dosage for individual de novo insertions, which is poorly estimated if low, the number of de novo insertions per explored haploid genome is estimated in an unbiased fashion. Let us call the number of distinct de novo insertions captured (at least once) $u$, and the total number of captured de novo insertions (i.e. accounting for repeated capture) $t$. The number of de novo insertions per explored haploid genome is $t/g$. It was 0.030, 0.019 and 0.024 for bulls 1, 2 and 3, respectively. This is the phenotype that was used for the GWAS (see above).

There is, however, information to determine at which time during development de novo insertions occur, in the frequency distribution of insertions that are re-captured 1, 2, 3, … times. If all observed de novo insertions are only captured once ($t = u$), it means that there is a large pool of very rare insertions that must all have occurred at a very late stage of spermatogenesis. If most de novo insertions are

recaptured multiple times ($t \gg u$), it means that de novo insertions occur at earlier stages of spermatogenesis. Thus, there is information about the developmental timing of de novo mobilization in the frequency distribution of the number of de novo insertions captured 1x, 2x, 3x, etc ($f_1, f_2, f_3, \ldots$). The corresponding distributions are shown in Supplementary Fig. 14 for the three bulls.

To gain insights in the developmental windows during which ERVK[2-1-LTR] mobilization takes place, we performed simulations under various scenarios to find those yielding recapturing frequency distributions that matched the real data best. We assumed a very simple model of gametogenesis in which 1,048,576 spermatogonial stem cells ($=2^{20}$) would derive from 20 successive binary cell divisions starting from a single precursor cell. A de novo insertion occurring at cell generation 1, would be characterized by a dosage per haploid genome (in sperm) of 0.5. One occurring at cell generation 10, would have a dosage of 0.00097, and at generation 21, a dosage of $4.8 \times 10^{-7}$. We defined windows of cell generations during which de novo mobilization may occur. Windows were centered around one specific generation (focal generation), that was most mobilization prone. Windows could be narrow (f.i. occurring in one generation only) or broad (up to 9 generations centered on the focal one). For windows encompassing multiple generations, the mutability of cells in a given generation was modelled using a binomial distribution, i.e., the mutability was highest for the focal generation and decreased with increasing distance from the focal generation. Given a choice of multi-generational window, the probability for a mobilization event to have occurred at one of the spanned generations was not only a function of the mutability of that generation, but also of the number of cells in that generation (which doubles at every additional generation). Having chosen a given window, we simulated de novo mobilization events, spread across the qualifying generations according to their relative probabilities, until the sum of the corresponding dosages corresponded to the number of de novo insertions per explored haploid genome (i.e. $t/g$) for the studied bull (1,2 or 3). Every in silico generated insertion corresponded to a ball in an urn with a sampling probability corresponding to its dosage. A ball corresponding to an insertion free genome was added to the urn with a sampling probability of $1-t/g$. We then sampled $g$ balls from this urn with replacement, using the corresponding vector of sampling probabilities, where $g$ is the actual number of studied haploid genomes for the three bulls (7690, 5386 and 9178). We then counted the number of non-insertion free balls (i.e. in silico insertions) that were sampled 1x, 2x, 3x, 4x and >4x. We performed 50 simulations per window and compiled the average number of observations for each category (1x, 2x, 3x, 4x and >4x). We compared this distribution with the corresponding real distribution (actual observations for bulls 1, 2 and 3). The quality of the match was quantified as the sum of the differences (absolute value) between simulated and real number of observations. Such simulations were conducted for window sizes ranging from 1 to 9, and sequentially considering every one of the 21 generations as focal one. Simulations were conducted separately by bull. As the results were very consistent for the three bulls, we summed the sum of the differences across bulls and selected the window that minimized the overall differences between simulated and real numbers across the three bulls (Supplementary Data 13). The corresponding window, as well as comparison between real and simulated frequency distributions, are shown in Fig. 8a, b.

### Simulating the evolution of ERV families in a panmictic population

We simulated the evolution of a panmictic population of constant size $N$ over $G$ generations. The next generation ($g_{i+1}$) was obtained from the previous one ($g_i$) by sampling $N$ times (with replacement) two parents (i.e. sex was not considered). The probability to become a

parent was affected by the fitness of the individual ($f_i$, range: 0 – 1). The fitness of the individual was determined by the total number of ERV elements in its genome ($t_i$), and was modelled as: $f_i = 1 - t_i^p / t_{max}^p$. $t_{max}$ is the maximum number of tolerated ERV elements in the genome, and was set at 250 (if $t_i \geq 250$, $f_i$ equals 0). The stringency of purifying selection was adjusted using $p$: $p = 1$ ("linear") > $p = 2$ ("quadratic") > $p = 3$ ("cubic"). In addition, we considered a scenario where fitness was not affected by the number of ERV elements ("none"; $f = 1$). At generation $g_1$, the population was "seeded" with one ERV element in the genome of a proportion $s$ of individuals. These initiating ERV elements were considered to be competent (i.e. drive mobilization by producing the needed machinery), and have an affinity ($a_{ij}$) of 1 for this machinery. Once the parents of an offspring were selected, we simulated Mendelian transmission of the parental ERV elements to the offspring assuming that each parental ERV has a 50% chance to be inherited by the offspring. In addition, we allowed for de novo ERV mobilization in the gametes. Every parental ERV had a certain probability ($q_{ij}$) to generate a new copy of itself in the gamete inherited by the offspring: $q_{ij} = r_i \times a_{ij} / \sum_{j=1}^{t_i} a_{ij}$. In this, $r_i$ is the de novo mobilization rate of parent $i$. It was determined by the number of competent ERV elements ($c_i$) in the parent's genome as follows: $r_i = (1 - e^{-0.1c_i}) \times r_{max}$, where $r_{max}$ is the highest possible mobilization rate set at 0.05. Once the offspring and its ERV elements were generated, the ERV elements were allowed to undergo mutation at a rate $\mu$. We tested 0.0001 and 0.001 as values for $\mu$. When an ERV element underwent mutation, (i) it would either become defective (if it was competent before) or stay defective (if it was already defective before), and – if the model allowed - (ii) its affinity ($a_{ij}$) might change (unless its affinity was already 0 before mutation). This was accomplished by sampling a $z$ value from $N(0,1)$ and adding it to the affinity of the ERV element prior to mutation. If the sum was negative, $a_{ij}$ was set at 0. The corresponding script was written in Perl and is made available in https://github.com/Lijingtangbo/TE-rate_manuscript.

**Reporting summary**

Further information on research design is available in the Nature Portfolio Reporting Summary linked to this article.

## Data availability

All WGS data of the 743 Dutch Holstein Friesian animals are deposited in European Nucleotide Archive under accession PRJEB53518. WGS data for 40 Belgian blue bulls are deposited in European Nucleotide Archive under accession PRJEB64406. PCIP sequencing data of 471 individual experiments generated in this study are available at European Nucleotide Archive under accession PRJEB64406. Complete sequences of the ERVK[2-1-LTR] inserted in *APOB*, five de novo insertions and 221 polymorphic ERVK[2-1-LTR]s are available from https://doi.org/10.5281/zenodo.7936181. The VCF file used for GWAS is available from https://doi.org/10.5281/zenodo.7937231. Source data are provided as a Source Data file. Source data are provided with this paper.

## Code availability

All scripts used in this study are available at GitHub (https://github.com/Lijingtangbo/TE-rate_manuscript) and at Zenodo (https://doi.org/10.5281/zenodo.10630335)[48–50].

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

## Acknowledgements

This project was conducted with funding from the ERC (ERC AdG-GA323030 to M.G., *Damona* project), the Walloon Region (DOG6 to M.G., *Causel* project), the University of Liège (FSR to C.C., *RetroBlue* project) and the Fund for Scientific Research in Belgium (F.R.S.-FNRS, PDR to C.C., *TE-rate* project); L.T. was supported by a Scholarship of the Chinese Scholarship Council (CSC); C.H.'s PhD fellowship was funded by Livestock Improvement Corporation (LIC, Hamilton, New Zealand); G.C.M.M. is post-doctoral research assistant of the H2020 EU project *BovReg* (GA815668); C.C. is senior research associate from the Fund for Scientific Research in Belgium (F.R.S.-FNRS). We are grateful to CRV (Arnhem, the Netherlands) for providing us with biological material for the *Damona* pedigree, to Inovéo (Ciney, Belgium) and FABROCA (Porcheresse, Belgium) artificial insemination centers for contributing Belgian blue sperm straws.

## Author contributions

C.C. and M.G. conceived and supervised the study; L.T. and S.D. performed the wet-laboratory experiments; L.T. and B.S. analysed the data and conducted all bioinformatics analyses; C.H. developed the LocaTER pipeline; G.C.M.M. mapped the WGS to the ARS-UCD1.2 reference genome; K.D. and M.A. gave technical support and conceptual advice; E.M. and A.S. provided pedigree data and biological material; W.C. and L.K. supervised sequencing data generation; M.G., C.C., L.T., and B.S. wrote the manuscript. All authors contributed to the final version of the paper.

## Competing interests

The authors declare no competing interests.
