## [Peer Review File · Nature Communications]

GWAS reveals determinants of mobilization rate and dynamics of an active endogenous retrovirus of cattleREVIEWER COMMENTS

Reviewer #1 (Remarks to the Author):

Overall, this is a technically super manuscript that conclusively shows there is an active, perhaps pathogenic, ERV in the germline of certain domestic bulls. They use a very interesting technique (CRISPR inverse PCR) to efficiently identify novel integrations in gametes, quantify the rate of de novo insertion per gamete, and characterize some integrations that persist in temporally isolated sperm cells- indicating the ERV is also integrating into genomic DNA in differentiating germ cells. The authors go on to sequence the proviruses they identified, and found that, although there is a significant number of intact (Type C) proviruses, most of the novel integrations are caused by an expansion of defective (Type D) proviruses. Furthermore, they find clear evidence of a higher mutational frequency in a CpG context (as expected) and APOBEC activity against incoming viruses.

Importantly, the authors did a GWAS study to find links between genetic factors and ERV-K2-1 expansion and identified a fully intact provirus on Chr19 to be the major driver. This would suggest that many of the novel integrations are the result of the activation and expression of this locus in these cattle.

Major Critiques:

1. As indicated in the second sentence of the abstract, the authors make the all-too-common mistake of conflating ERVs with retrotransposons, apparently assuming, without evidence, that once a provirus enters the germline, it immediately becomes capable of retrotransposition within the germline. The fact is that retroviruses, including infectious ERVs, have a number of characteristic features of their replication cycle that are incompatible with a switch to an intracellular transposition. The authors use the term retrotransposon throughout the paper while discussing their results, and build their modeling based on the same assumption.

While such a conversion has happened with a couple of mouse elements, these tend to have divergent MA signals and lack envelope genes and they likely required a long period of evolution to arise and spread. The sequences they share for this bovine ERV look to have a clear MA myristylation signal and a full-length, likely functional, envelope gene, which wouldn't fit with a retrotransposon. It's far more likely that these elements arose from viral reinfection of the germ line, probably mediated by expression of the intact endogenous elements, particularly the one on Chr19, in the genome.

The distinction between retrotransposition and reinfection is biologically quite important, because the activation and production of the infecting virus could (and probably did) occur in some nearby, non-germline cell, perhaps a tubular epithelial, blood, or Sertoli cell, after which the released virus made its way to and infected a germline cell by the usual, Env-requiring pathway.

Some might argue that the viral mechanism could also be called "retrotransposition," because the outcome is the same, but that broad use of the term has such a large power to mislead, because the root "transpose" clearly implies an intracellular mechanism.

As noted below, the authors do point out, in brief passages here and there, some of these issues, leading to a sense of cognitive dissonance, when they ignore the implications of what they write. This is a technically beautiful study, but I would not consider it acceptable until the authors rewrite to remove the almost certainly incorrect transposon virus from their interpretations. If it were me, I would refer to the process as reinfection," but a more neutral "mobilization" throughout would also be acceptable, with a balanced presentation of the different possible mechanisms in the discussion.

2. The paper models the spread of the defective virus quite well. They mention examples of "parasite of a parasite" ERVs in koalas and mice, but not humans. Indeed, this is a retroviral phenomenon that has been known with MLV for more than 40 years (look up "VL30"). It would also be good to mention that humans also have a defective (Type 1) ERV-K that looks to be carried along by (and recombine with) the full length (Type 2) proviruses.

3. The manuscript (Line 102) identifies the proviruses of interest as belonging to the "[2-1-LTR] subfamily of ERVK elements," without a reference where this group is defined or even what genus it belongs to. Full understanding of the implications of the results would be greatly aided by a supplemental figure showing how the phylogenetic relationship of the intact provirus to the rest of Retroviridae, or at least inclusion of a reference that does this. Further, "subfamily" is just careless. All retroviruses except spumaviruses belong to the same subfamily. Please don't use taxonomic terminology loosely. Better is to use some neutral word, like "group" devoid of taxonomic

implication. Finally (Line 133ff), "ERVK" proviruses are defined by the imputed (lysine) tRNA primer and can belong to otherwise relatively unrelated groups. Is that how they are identified here? Or was it done by sequence similarity to a canonical (e.g., HERV-K) provirus?

4. It would be of considerable interest to know how the ERV group identified in the animals studied here is distributed among other cattle and relatives. Probably a topic for another paper, but any information the authors have at this time would be welcome here.

Other points:

Line 55 (and elsewhere): "Halve". Should be "half." Odd error, considering that the English is generally quite good.

Line 77-78 "...thought to derive..." just "derived" will do. There is no question about it.

Line 308: '+' is undefined.

Line 535: "stem," not "stems."

Line 561-562: The most likely explanation is difference in the level of expression. In the virus producing cell.

Line 943: "dosage of ..."?

Line 579-581. As discussed above, the hidden assumptions underlying this statement—that proviruses automatically become capable of intracellular transposition once they enter the germline, and that the exogenous virus that gives rise to the ERV automatically disappears once it does so – are popular, but completely groundless. In the opinion of this reviewer, they greatly reduce the potential of this paper to make a significant contribution to the ERV literature. The authors should recast their approach to the far more likely explanation that the mobilization of the elements studied follows expression and cell-cell transmission of the ERV-encoded virus.

Reviewer #2 (Remarks to the Author):

I have read the manuscript titled "A bovine GWAS reveals determinants of mobilization rate and dynamics of endogenous retroviruses". I am a researcher with a decade of experience working on transposable elements and their silencing mechanisms. Here are a few comments on the manuscript which in my opinion is high quality and interesting work.

The authors create a full genome DNA sequence dataset in cattle affected with a new autosomal disorder (cholesterol deficiency), known to be caused by a new transposable element insertion at the APOB locus. This is a rare opportunity as they have the full lineage of these animals, which they couple with their NGS sequencing data to find new transposable element integration events. Their analysis seeks to investigate replication dynamics and factors influencing the rate of transposition. Their results could not find an association with genomic factors outside of transposons affecting rate of transposition, hinting that repressive mechanisms to target this new burst of transposition activity have not evolved yet – this is itself a strong finding with implications of our understanding of how and when the mechanisms involved in repression arise. They find an association suggesting that the accumulation of defective integrants of transposons of the same family play a role in competitively dampening the ability of full-length elements to transpose. The manuscript offers a rare window into dynamics of transposable elements and should be interesting to people in the mobile DNA field of study, as well as many in the veterinary medicine field.

Comments:

1. As it was found that "three of the four male mobilizations occurred in the germline of the same bull, of which two in the same sperm cell", is there a way to investigate, or speculate, on a possible loss of transposable element control in that original male animal?
2. PCIP-Seq - add a small sentence discussing that the technique is also blind if mutations are found in the CRISPR target sites - speculate how that might affect some of the results and discussion.
3. Mention how the common (frequency ≥ 0.05) polymorphic ERVK[2-1-LTR] elements are distributed in the genome in the cohort of animals - are some individual elements more frequent than others, any evidence of selection (outside the artificial selection imposed by the use of the original carrier animal)? Are they close to genes / pathways, more / less distributed as the other elements of the family are (intergenic vs proximity to genes)?

4. Figure 2b - The 'genome' label is confusing, and the comparison should be made for all ERV type subcategories.
5. Figure 4c - any genes in the vicinity? If so, please add a gene track schematic underneath each plot.
6. Figure 7a is partially unclear to me - it might be best to represent the whole data as a heatmap of all observed combinations since it maxes out at 26 rows.
7. Figure 7b, c and d - please order the columns in the same way between the 3 plots, it is hard to compare between them in the current state.

Minor points:

1. Please provide if possible as a supplementary figure a phylogeny (maybe as a circular plot) of all elements, based on multiple alignment. Metadata annotation around the plot would be insightful (allele frequency, category of recombinants, etc).
2. The tone of language through the manuscript when addressing transposable elements is overly negative. While I understand that the author's starting point is a new transposon causing disease, it is important not to skew readers perception to their positive potential. Examples:
 "This mobilization of ERVs is a source of mostly deleterious variation"
 "ERV endogenization has been caught in the act"
 "We propose that D-type elements act as parasite-of-parasite"
 Please update the text for a more balanced view of transposons, which should be more neutral (as most elements are) and not discount that they are important drivers of innovation.
3. Some precision in the language in some parts - for example:
 "As an example, the syncytin genes, which are essential for placentation, are thought to derive from ancient ERVs"
 Syncytin origin being an Env gene from an ancient ERV is well established, so "thought to" is imprecise
4. Figure 2c undersells the size of the dataset - at first glance it looks like the whole pedigree is a few animals, and could mislead some readers. Please visually update, or add some small text on the figure panel with the number of animals at each level
5. "Five to ten percent of mammalian genomes is occupied by multiple families of endogenous retroviruses (ERVs), which may count thousands of dispersed members" - change to 'each of which may count'
6. "ERVs, leaving only "DNA fossils" attesting" - drop 'only'
7. "Halve of mammalian genomes" - "Half"

Reviewer #3 (Remarks to the Author):

Dear Authors,

First of all I have to apologize for my English. I am not practicing this language too often so do not expect smooth English or perfect use of words meanings or so. I try to do my best.

Before going to the review itself I have to admit that in my 35 year scientific life your paper was the most complicated, complex and demanding careful reading. Moreover, reading it was a pleasure because I felt page after page that I myself understand a new and fascinating phenomenon which I never was conscious.

Being captured by these feeling, I did not give up and tried to express my comments which I believe might be considered by Authors as having some value. They are written in spirit of sharing an independent view to what is the subject of the manuscript.

General remarks:

1. The title does not reflect exactly the results since the described phenomenon occurs primarily in Belgian Blue breed (although Holstein bulls were also used especially in population genetics analysis). So would suggest to extend the title, by adding at the end "...in Belgian Blue and

Holstein bulls". I can admit that the problem is probably occurring in other breeds but the manuscript describes what happened in BB HOL breeds.

2. I would appreciate very much to insert into MM section what was the genotype of Myostatin causing Muscle Hypertrophy - do they have the same genotype or different, and if different what was the distribution of genotypes.

3. If I am not wrong, Belgian Blue cattle, and especially breeding bulls can be substantially related to each other (inbred). Therefore I would suggest to put inbreeding coefficient IC data into MM section. Moreover, if there are significant differences in IC within the bull population studied, I would suggest to take into account this variable into any statistical model used in this work to better recognize the factors influencing mobilisation rate and dynamics of endogenous retroviruses.

4. From the same reason expressed in point 3, I would suggest to include into statistical model used throughout the work, the Estimated Breeding Value of bull (conventional or genomic) since this can also potentially be a factor somehow influencing mobilisation rate and dynamics of endogenous retroviruses.

5. Almost all figures in main manuscript (and also all in Extended data) are very dense in information which is very positive, but any reader expects that all images tables or drawings must be visible enough for normal reading in printed A4 page. That is often impossible, especially Figure 3, 4 and 5. I would suggest to give one page for the content of the figure, and transfer the Legend for next page.

Specific comments:

Line 134 - explain shortly what is the difference between ERVK and ERVL

Line 136-137 - what statistical model generated this value?

Line 222 - SS shortcut should be explained here not later (line 228)

Line 182 - explain in more detail understanding of "shared shearing sites"

Line 211 - "more often" means statistically verified?

Line 246 - put the method or software (and Reference) used for imputation of genotypes

Line 253 - I would like to see the SNP cluster image, its quality (add to Supplementary section)

Line 252 - I again suggest to enrich the linear model used in GWAS with for me obvious variables, eg. relatedness matrix, sire effect, EBV

Line 591 - put the Reference - who is the author of "parasite-of-parasite" concept and explain in few sentences this concept.

Point-by-point response to the reviewers' comments

We thank the reviewers for their very helpful comments, which we have addresses as follows:

REVIEWER COMMENTS

Reviewer #1 (Remarks to the Author):

Overall, this is a technically super manuscript that conclusively shows there is an active, perhaps pathogenic, ERV in the germline of certain domestic bulls.

We thank the reviewer for this very positive statement.

They use a very interesting technique (CRISPR inverse PCR) to efficiently identify novel integrations in gametes, quantify the rate of de novo insertion per gamete, and characterize some integrations that persist in temporally isolated sperm cells- indicating the ERV is also integrating into genomic DNA in differentiating germ cells.

The authors go on to sequence the proviruses they identified, and found that, although there is a significant number of intact (Type C) proviruses, most of the novel integrations are caused by an expansion of defective (Type D) proviruses. Furthermore, they find clear evidence of a higher mutational frequency in a CpG context (as expected) and APOBEC activity against incoming viruses.

Importantly, the authors did a GWAS study to find links between genetic factors and ERV-K2-1 expansion and identified a fully intact provirus on Chr19 to be the major driver. This would suggest that many of the novel integrations are the result of the activation and expression of this locus in these cattle.

Major Critiques:

1. As indicated in the second sentence of the abstract, the authors make the all-too-common mistake of conflating ERVs with retrotransposons, apparently assuming, without evidence, that once a provirus enter the germline, in immediately becomes capable of retrotransposition within the germline. The fact is that retroviruses, including infectious ERVs, have a number of characteristic features of their replication cycle that are incompatible with a switch to an intracellular transposition. The authors use the term retrotransposon throughout the paper while discussing their results, and build their modeling based on the same assumption. While such a conversion has happened with a couple of mouse elements, these tend to have divergent MA signals and lack envelope genes and they likely required a long period of evolution to arise and spread. The sequences they share for this bovine ERV looks to have a clear MA myristylation signal and a full-length, likely functional, envelope gene, which wouldn't fit with a retrotransposon. It's far more likely that these elements arose from viral reinfection of the germ line, probably mediated by expression of the intact endogenous elements, particularly the one on Chr19, in the genome.

The distinction between retrotransposition and reinfection is biologically quite important, because the activation and production of the infecting virus could (and probably did) occur in some nearby, non-germline cell, perhaps a tubular epithelial, blood, or Sertoli cell, after which the released virus made its way to and infected a germline cell by the usual, Env-requiring pathway.

Some might argue that the viral mechanism could also be called "retrotransposition," because the outcome is the same, but that broad use of the term has such a large power to mislead, because the root "transpose" clearly implies an intracellular mechanism.

As noted below, the authors do point out, in brief passages here and there, some of these issues, leading to a sense of cognitive dissonance, when they ignore the implications of what they write.

This is a technically beautiful study, but I would not consider it acceptable until the authors rewrite to remove the almost certainly incorrect transposon virus from their interpretations. If it were me, I would refer to the process as reinfection," but a more neutral "mobilization" throughout would also be acceptable, with a balanced presentation of the different possible mechanisms in the discussion.

Response 1.1

We thank the reviewer for this comment/suggestion, with which we agree. We have extensively reworked the manuscript (and especially the Discussion) to clarify interpretation and description of our findings. Specifically:

- **Abstract:** We now readily mention reinfection and retro-transposition as possible amplification mechanisms of ERV clades: (lines 33-35) *“New ERV clades arise by retroviral infection of the germline followed by expansion as endogenized proviruses generate copies of themselves by reinfection and/or retrotransposition.”*
- **Introduction:** The two steps referred to by the reviewer were already mentioned in the introduction. We hope that the present phrasing will be clear for the readers: (lines 56-64): *“ERVs derive from retroviral infection of the germline enabling vertical viral transmission. Such retroviral endogenization may lead to the progressive expansion of a clade of ERV elements that may count tens to thousands of members, by an ERV-encoded reverse transcriptase (RT)-dependent copy-paste mechanism. At first, this process entails ERV-encoded envelope (ENV)-dependent budding of viral particles followed by germline reinfection. Subsequently, the ERV may at least in part forgo the extracellular phase yet continue to multiply by more efficient intracellular retro-transposition. In mice, such a transition from IAPE to IAP elements has been shown to result from the combined acquisition of a novel GAG addressing signal and loss of ENV.”*
- **Discussion:** We have added a section emphasizing the fact that the ERTVK[2-1-LTR] elements most likely mobilize by reinfection: (lines 625-633) *“The observation of an intact ENV ORF in a large proportion of C-type elements, combined with in silico prediction that the matrix domain (MA) of the GAG protein of C-type elements is a target for N-terminal myristoylation (essential to address viral particles of both exogenous and endogenous retroviruses to the cell membrane; Extended data Fig. 7), strongly suggests that ERVK[2-1-LTR] elements still multiply by within-host, intercellular reinfection rather than by the supposedly more effective intracellular retro-transposition route. Which cells produce the viral-like particles (somatic or germline cells), which membrane receptor viral-like particles recognize in the recipient cells, and which reinfection path is used (free particles, virological synapses or microtubule-mediated transport) remains unknown.”*

2. The paper models the spread of the defective virus quite well. They mention examples of “parasite of a parasite” ERVs in koalas and mice, but not humans. Indeed, this is a retroviral phenomenon that has been known with MLV for more than 40 years (look up “VL30”). It would also be good to mention that humans also have a defective (Type 1) ERV-K that looks to be carried along by (and recombine with) the full length (Type 2) proviruses.

Response 1.2

The reviewer is correct to highlight that trans complementation between mobile elements and viruses has been extensively documented before and that this was not sufficiently acknowledged in the previous version of the manuscript. What we were mainly trying to highlight, however, is the remarkable similarity, in terms of genome structure, between the C/D pair, and two heavily studied (within clade) “master-slave” pairs: MusD/ETn and KoRV A/RecKoRV, and the fact that this may point towards shared cis-effects.

We have reworked the **discussion** to clarify these points (lines 665-674): “*Trans-complementation between mobile elements has been extensively documented before* ^[26]. *Examples of complementation in trans between morphs of the same ERV clade include ETn by MusD* ^[31,32] *and IAP IΔ1 by IAP* ^[33,34] *in mice, RecKoRV by KoRV A in koala* ^[35,36], *and possibly Type 1 by Type 2 HERV-K in human* ^[37]. *Strikingly, the bovine ERVK[2-1-LTR] C/D pair, murine MusD/ETn pair and koala KoRV A/RecKoRV pair share a pattern of swapping of a central segment of the competent ERV element with an old piece of retroelement, yet conservation of flanking sequences encompassing portions of the GAG and ENV (ERVK[2-1-LTR] and KoRVA) or GAG and POL (MusD) genes (in addition to the LTRs), which may be suggestive of cis-effects on the effectiveness of trans-complementation as reported for IAP IΔ1* ^[33].”

Thus, we are now referring to the likely Type 1 by Type 2 HERVK trans-complementation as recommended by the reviewer. Although we acknowledge the VL30 by MLV trans-complementation, we do not mention it in this discussion as we aimed to highlight examples of trans-complementation between “morphs” belonging to the same clade. We hope that the reviewer will be satisfied with this distinction.

3. The manuscript (Line 102) identifies the proviruses of interest as belonging to the ‘[2-1-LTR] subfamily of ERVK elements,’ without a reference where this group is defined or even what genus it belongs to. Full understanding of the implications of the results would be greatly aided by a supplemental figure showing how the phylogenetic relationship of the intact provirus to the rest of Retroviridae, or at least inclusion of a reference that does this. Further. “subfamily” is just careless. All retroviruses except spumaviruses belong to the same subfamily. Please don’t use taxonomic terminology loosely. Better is to use some neutral word, like “group” devoid of taxonomic implication. Finally (Line 133ff), “ERVK” proviruses are defined by the imputed (lysine) tRNA primer and can belong to otherwise relatively unrelated groups. Is that how they are identified here? Or was it done by sequence similarity to a canonical (e.g., HERV-K) provirus?

Response 1.3

We thank the reviewer for these recommendations. We have added a section describing the relationship with other retroviridae, the likely functionality of the ENV and GAG proteins supporting mobilization by inter-cellular reinfection, and the complementarity of the consensus binding sites with the 3’ extremities of bovine lysine tRNAs in **Results** with an accompanying Extended data Fig. 7: (lines 351-362) “*The consensus GAG, POL (reverse transcriptase domain) and ENV (transmembrane domain) sequences of C-type elements each cluster with betaretroviridae (Extended data Fig. 7a). The ENV protein encompasses a surface unit (SU), proteolytic processing site, and transmembrane unit (TM) with fusion peptide, CX₇C cysteine motif, transmembrane region (TR) and cytoplasmic tail (CT) domains, typical of beta- and lenti-retroviruses* ^[23]. *The amino-terminal end of the GAG protein was predicted by Myristoylator* ^[24] *to correspond to a myristoylation site (high confidence score of 0.98) (Extended data Fig. 7b). Taken together this suggests that ERVK[2-1-LTR] are able to mobilize by inter-cellular reinfection* ^[25]. *The consensus primer binding site (PBS), shared by C-type and D-type elements (Suppl. Table 8), presents striking complementarities with the 3’ extremities of the two bovine lysine tRNAs (CTT and TTT codons, respectively), supporting the ERVK denomination (Extended data Fig. 7c).”*

Extended data: Figure 7

b

MMTV	M	G	V	S	G	S	K	G	Q	K	L	F	V	S	V	L	Q	R	L	L	-	-	0.9753
MPMV	M	G	Q	E	L	S	Q	-	H	E	R	Y	V	E	Q	L	K	Q	A	L	K	-	0.9839
JSRV	M	G	Q	T	H	S	-	R	Q	L	F	V	H	M	L	S	V	M	L	K	H	-	0.9781
ERVK2-1-LTR	M	G	N	T	E	S	N	E	R	Q	L	F	I	G	V	I	L	Q	L	L	-	-	0.9838

(a) Unrooted phylogenetics trees of GAG, POL (reverse transcriptase domain) and ENV (transmembrane domain) sequences of ERVK[2-1-LTR] and representative Alpha, Beta, Gamma, Delta, Epsilon, Lenti and Spuma exogenous retroviruses. ALV: Avian leukosis virus; RSV: Rous sarcoma virus; MMTV: Mouse mammary tumor virus; MPMV: Mason-Pfizer monkey virus; JSRV: Jaagsiekte sheep retrovirus; HTLV-1: Human T-lymphotropic virus 1; BLV: Bovine leukemia virus; WDSV: Walleye dermal sarcoma virus; MLV: Moloney murine leukemia virus; FLV: Feline leukemia virus; HIV-1: Human immunodeficiency virus 1; BIV: Bovine immunodeficiency virus; BFV: Bovine foamy virus; EFV: Equine foamy virus.
(b) Sequence of the N-terminal domain of the representative beta retroviruses and ERVK[2-1-LTR] Gag protein disclosing a consensus sequence required for myristoylation (M)GXXXS/T with the first M corresponding to the Gag initiation codon and a domain rich in positively charged basic residues (basic domain in red) interacting with the membrane phospholipids. The score for myristoylation was predicted by Myristoylator (<https://web.expasy.org/cgi-bin/myristoylator>).
(c) Consensus primer binding site (PBS) shared by C and D-type elements aligned to the 3' end of the bovine CTT and TTT lysine tRNA genes.

We have changed the term “subfamily” to “clade” throughout the manuscript when referring to ERVK[2-1-LTR], and “subclade” or “morph” when referring to the C- and D-type elements, as well as solo-LTRs.

4. It would be of considerable interest to know how the ERV group identified in the animals studied here is distributed among other cattle and relatives. Probably a topic for another paper, but any information the authors have at this time would be welcome here.

Response 1.4

We have added an entire section on this topic in the **Discussion**: (lines 575-612) “**The ERVK[2-1-LTR] clade derives from ancient endogenization of an unknown exogenous retrovirus. That a high proportion of ERVK[2-1-LTR] loci are polymorphic suggests that their colonization of the bovine genome is a relatively recent event. However, contrary to recent episodes of ERV endogenization reported in poultry, mice, sheep/goat, koala and cat [5], the identity of the exogenous retrovirus at the origin of the bovine ERVK[2-1-LTR] family remains**

unknown and may be extinct. Sequences with similarities $\geq 95\%$ over $\geq 95\%$ of the full length of the chromosome 19 C-type element exist in the reference genomes of other domesticated taurine (*Bos taurus*) and indicine breeds (*Bos indicus*), gaur (*Bos gaurus*), gayal (*Bos frontalis*), domestic yak (*Bos grunniens*), wild yak (*Bos mutus*), bison (*Bison bison*), water buffalo (*Bubalus bubalis*), and African buffalo (*Syncerus caffer*). A marked drop in similarity is observed when querying the genomes of caprinae, including sheep (*Ovis aries*) and goat (*Capra hircus*) (Suppl. Table 14). This suggests that ERVK[2-1-LTR] endogenization may have occurred in an ancestor of Bovinae, i.e. ~ 15 million years ago. To gain additional insights in the history of the ERVK[2-1-LTR] clade we compared the shared GAG and ENV sequences between C- and D-type elements. We restricted the analysis to regions immediately flanking the boundaries of the C- and D-specific segments and presenting little or no evidence of recombination. The average pairwise difference between C- and D-type elements was 38.4 in 250 base-pairs. Assuming a de novo mutation rate of 1×10^{-8} base pairs per generation^[28], this would correspond to ~ 15 million generations or ~ 45 million years. It is, however, possible that some of the differences between C and D-type shared sequences were introduced upon creation of D-type elements, and/or that the mutation rate for ERV elements, influenced by the retro-transposition process, is higher than 1×10^{-8} base pairs per generation (Suppl. Fig. 2). Anyhow, these analyses corroborate the prediction that the initial ERVK[2-1-LTR] endogenization occurred prior to the divergence of boviniae. We also analyzed the sequence divergence between the 5' and 3' LTRs of all C and D-type elements. Upon creation of a new insertion, 5' and 3' LTR have identical sequence. Observed differences therefore reflect the accumulation of de novo mutation and therefore presumably provide information about the time of insertion. For 188 of the 220 elements with complete sequence information, 5' and 3' LTR were identical over their entire 1,287 base pair length (Suppl. Table 8). Assuming a mutation rate of 1×10^{-8} base pairs per generation (and that all elements inserted at the same time), this amounts to an estimated insertion time point (t) of $\sim 6,000$ generations or $\sim 18,000$ years ago ($t = \log(188/220) / \log((1 - 10^{-8})^{2574})$). If we restrict the analysis to the (older) elements with population frequency ≥ 0.10 , the estimates of t increase to $\sim 15,000$ generations or $\sim 45,000$ years ago. These estimates are dramatically different from the ones obtained from the phylogenetic distribution of ERVK[2-1-LTR] elements and from the comparison of the C- and D- shared sequences. We believe that they actually inform about the time it takes for a newly inserted element to undergo intramolecular recombination and be converted to a solo-LTR element, and not about the actual time of insertion.”

Other points:

Line 55 (and elsewhere): “Halve”. Should be “half.” Odd error, considering that the English is generally quite good.

1.5. Has been corrected.

Line 77-78 “...thought to derive...” just “derived” will do. There is no question about it.

1.6. Has been corrected.

Line 308: ‘+’ is undefined.

1.7. We are now defining the meaning of “+” (line 318-320): *“Examination of the whole genome sequences (WGS) of the 40 bulls, revealed that this was due to the segregation of a third solo-LTR allele (in addition to “+” and full-length ERV, where “+” corresponds to the ancestral allele) for these loci.”*

Line 535: “stem,” not “stems.”

1.8. Has been corrected.

Line 561-562: The most likely explanation is difference in the level of expression. In the virus producing cell.

1.9. The possible causes of the preferential mobilization of D-type elements are now briefly addressed in the Discussion (lines 661-665): *“That D-type insertions are able to outnumber C-type insertions could either be due to the fact that D-type gRNA are more abundant in cells than their C-type counterparts (f.i. because some of them are transcribed at a higher rate or are more stable), or because they are more effective in utilizing the C-type provided machinery to generate ecDNA^[30].”*

Line 943: “dosage of ...”?

1.10. Is now specified in the text (lines 923-925): *“In a second round of GWAS, dosage of the chromosome 19 ERVK[2-1-LTR] locus (i.e. number of alleles with the ERVK[2-1-LTR] insertion) was added as an extra covariate in the model.”*

Line 579-581. As discussed above, the hidden assumptions underlying this statement—that proviruses automatically become capable of intracellular transposition once they enter the germline, and that the exogenous virus that gives rise to the ERV automatically disappears once it does so – are popular, but completely groundless. In the opinion of this reviewer, they greatly reduce the potential of this paper to make a significant contribution to the ERV literature. The authors should recast their approach to the far more likely explanation that the mobilization of the elements studied follows expression and cell-cell transmission of the ERV-encoded virus.

We thank the reviewer for this comment/suggestion, with which we agree. We have extensively reworked the manuscript (and especially the Discussion) to clarify interpretation and description of our findings. See Response 1.1 for specifics.

Reviewer #2 (Remarks to the Author):

I have read the manuscript titled "A bovine GWAS reveals determinants of mobilization rate and dynamics of endogenous retroviruses". I am a researcher with a decade of experience working on transposable elements and their silencing mechanisms. Here are a few comments on the manuscript which in my opinion is high quality and interesting work.

The authors create a full genome DNA sequence dataset in cattle affected with a new autosomal disorder (cholesterol deficiency), known to be caused by a new transposable element insertion at the APOB locus. This is a rare opportunity as they have the full lineage of these animals, which they couple with their NGS sequencing data to find new transposable element integration events. Their analysis seeks to investigate replication dynamics and factors influencing the rate of transposition. Their results could not find an association with genomic factors outside of transposons affecting rate of transposition, hinting that repressive mechanisms to target this new burst of transposition activity have not evolved yet – this is itself a strong finding with implications of our understanding of how and when the mechanisms involved in repression arise.

They find an association suggesting that the accumulation of defective integrants of transposons of the same family play a role in competitively dampening the ability of full-length elements to transpose. The manuscript offers a rare window into dynamics of transposable elements and should be interesting to people in the mobile DNA field of study, as well as many in the veterinary medicine field.

We thank the reviewer for this positive comment.

Comments:

1. As it was found that "three of the four male mobilizations occurred in the germline of the same bull, of which two in the same sperm cell", is there a way to investigate, or speculate, on a possible loss of transposable element control in that original male animal?

Response 2.1

We have mined the whole genome sequence of the corresponding bull for evidence of loss-of-function mutations in 38 genes that – based on available literature - may play a role in controlling ERV mobilization. We found no evidence for such occurrence. This is now mentioned in the **Results** section (lines 159-162): *“Mining of the whole genome sequence of the bull transmitting three ERVK[2-1-LTR] de novo insertions did not reveal striking anomalies in 38 genes that have been connected with control of ERV mobilization^[16] (Suppl. Table 2). ”*

2. PCIP-Seq - add a small sentence discussing that the technique is also blind if mutations are found in the CRISPR target sites - speculate how that might affect some of the results and discussion.

Response 2.2

This is correct. We have added a sentence in the **Results** section to highlight this (lines 200-202): *“Of note, ERVK[2-1-LTR] with polymorphisms in the CRISPR target sites may escape detection by PCIP, an issue which is at least partially mitigated by targeting both 5' and 3' LTR.”*.

We were aware of this issue. This is the reason why we mention in the **Results** section (lines 323-325): *“The presumed solo-LTR status was confirmed by PCR for 84 tested elements, indicating that no important ERVK[2-1-LTR] subclade (other than C and D) was missed by PCIP (Suppl. Table 9).”*.

3. Mention how the common (frequency ≥ 0.05) polymorphic ERVK[2-1-LTR] elements are distributed in the genome in the cohort of animals - are some individual elements more frequent than others, any evidence of selection (outside the artificial selection imposed by the use of the original carrier animal)? Are they close to genes / pathways, more / less distributed as the other elements of the family are (intergenic vs proximity to genes)?

Response 2.3

- We have examined the genomic distribution of ERVK[2-1-LTR] elements stratified by frequency: common ones ($MAF \geq 0.05$) and rare ones ($MAF < 0.05$). There was no significant difference between the two groups ($p = 0.95$).

- We have analyzed the distribution of allelic frequencies for loci with solo-LTRs, C-type and D-type allele, as well as the small numbers of tri-allelic loci. The results are reported in Suppl. Fig. 4, and referred to in the **Results** section: (lines 325-327) *“The distribution of allele frequencies in the 40 sequenced bulls was slightly shifted towards higher values for solo-LTRs when compared to C- and D-type ($p = 1.4 \times 10^{-4}$ and 2.0×10^{-8}), supporting their older age as expected (Suppl. Fig. 4).”*, as well as in the **Discussion**: (lines 564-574) *“We show that ERVK[2-1-LTR] elements come in three “morphs”: solo-LTR, C(ompetent)-type elements and D(effective)-type elements. The majority of loci are biallelic, i.e. characterized by the wild-type “+” allele and one “ERV” allele which can be of C- (~9% of loci), D- (~47%), or solo-LTR-type (~41%). A minority of loci are triallelic (~2%), characterized by the cosegregation of “+” allele, a full-length (C- or D-type) ERV allele, and the derived solo-LTR allele (Suppl. Fig. 4). Of note, solo-LTRs are expected to be older than their full-length counterparts. Accordingly, their frequency spectrum is shifted upwards. The estimated proportions of C-, D-, and solo-LTR-type loci therefore change with sample size: as sample size increases, more new C- and D-type than solo-LTR-type loci are uncovered, hence their proportion is increasing at the expense of the solo-LTR class.”*

Supplementary Figure 4

Distribution of allelic frequencies of ERVK[2-1-LTR] segregating in Belgian Blue cattle.

ERVK[2-1-LTR] elements are sorted by "morph": C-type, D-type, solo-LTR. Within morph, the ERVK[2-1-LTR] elements are ranked by their frequency in the Belgian Blue population. At five loci, solo-LTR, D-type and wild-type (+, grey) allele coexist at the shown frequencies (M(ultiple) morph).

- We have performed the Extended Haplotype Homozygosity following Voigt et al. (2006) and using the rehh R package (Klassmann & Gautier, 2022) to search for signatures of positive selection (hard sweeps). We did not observe convincing evidence for significant departures from expectations under H₀. The following figure illustrates the results obtained for the Chr 19 locus:

- We took all genes mapping within one megabase of all ERVK[2-1-LTR] elements with population frequency ≥ 0.05 , and subjected the ensuing list to Reactome analysis (<https://reactome.org>). There was no convincing evidence for significantly enriched pathways.

4. Figure 2b - The 'genome' label is confusing, and the comparison should be made for all ERV type subcategories.

Response 2.4

We have replaced the labels of Fig. 2b, which now read "Expected" and "Observed".

We have repeated the analysis for ERVK and other ERV elements separately. The results are reported in **Results:** (lines 145-146) "These trends did not differ significantly between the ERVK and non ERVK groups ($p_{genic\ vs\ intergenic} = 0.61$; $p_{sense\ vs\ antisense} = 0.42$) (Extended data Fig. 2b-c)." and accompanying figure:

Extended data: Figure 2

(a) Proportional representation of ERVK sub-groups in genome space (light grey bars) and amongst polymorphic elements detected by Locater (dark grey bars). Repbase reports 33 subgroups of ERVK, of which the most abundant in the reference genome are BTLTR1 (38.9%), ERVK[2-1-LTR] (20.2%), BLTR1B (10.2%) and BLTR1J (7.1%). While BTLTR1 is underrepresented amongst polymorphic ERVK elements (30.1%), ERVK[2-1-LTR] (26.6%), BLTR1B (27.1%) and BLTR1J (8.8%) are respectively overrepresented 1.3, 2.6 and 1.2 times. Of note, the very rare ERVK[2-3-LTR] subgroup (0.5%) is 4 times overrepresented amongst polymorphic ERVK elements. This suggests that the latter four subgroups, especially, might still be active.

(b) Genomic distribution of polymorphic ERVK elements (dark grey) compared to the corresponding genome space (light grey). Left: intergenic versus genic space. Right: antisense versus sense orientation for genic elements.

(c) Genomic distribution of polymorphic ERV elements other than the ERVK group (dark grey) compared to the corresponding genome space (light grey). Left: intergenic versus genic space. Right: antisense versus sense orientation for genic elements. The respective proportions did not differ significantly between ERVK and non-ERVK elements. See also main text.

(d) Derived Allele Frequency (DAF) spectrum for genic (dark grey) and intergenic (light grey) ERV elements. DAF of genic insertions are slightly ($p = 0.06$) shifted towards lower values as expected under purifying selection.

(e) Derived Allele Frequency (DAF) spectrum for genic antisense (dark grey) and genic sense (light grey) ERV elements. There is no significant evidence ($p = 0.99$) for a shift of DAF of sense insertions towards lower values when compared to antisense, as expected if they would be subject to stronger purifying selection.

5. Figure 4c - any genes in the vicinity? If so, please add a gene track schematic underneath each plot.

Response 2.5

We have added a track with gene content in Fig. 4c, akin to what we did in Extended data 5.a.

Figure 4

GWAS for the rate of de novo ERVK[2-1-LTR] mobilization in the male germline of cattle.

(a) GWAS conducted using PCIP-determined mobilization rate in sperm samples of 430 Belgian Blue bulls and genotypes at ~10 million variants, revealing a very strong signal on chromosome 19. (b) GWAS conducted using the same population after correcting ERVK[2-1-LTR] mobilization rate for the effect of the chromosome 19 QTL. Seven additional (near) genome-wide significant effects were detected. Significant loci encompassing an ERVK[2-1-LTR] element are highlighted by red dots, others by light blue dots. (c) Zoom into the four loci encompassing an ERVK[2-1-LTR] element, shown as triangles (as opposed to circles for SNPs). Variants are colored according to their LD (r^2) with the lead variant. The LD between the ERVK[2-1-LTR] element and the lead SNP was 1.00 (ERV = lead variant) for chromosome 19, 0.62 for chromosome 2, 0.85 for chromosome 4, and 0.81 for chromosome 21.

6. Figure 7a is partially unclear to me - it might be best to represent the whole data as a heatmap of all observed combinations since it maxes out at 26 rows.

Response 2.6

We have reworked the corresponding part of the legend to Figure 7 to hopefully clarify the message:

“Figure 7: Preferential mobilization of D-type ERVK[2-1-LTR] elements. (a) Position of the four PCIP-tags in the ERVK[2-1-LTR] genome. Each tag is characterized by a number of distinct variants across the 309 ERVK[2-1-LTR] elements segregating in Belgian Blue cattle, ranging from 5 (tag 2) to 26 (tag 1). The 309 ERVK[2-1-LTR] elements are each characterized by (at least) one combination of variants, i.e. a haplotype. We observed 77 distinct haplotypes in the BB population (which we refer to as haplotypes A1 to A77), indicating that different ERVK[2-1-LTR] elements share the same haplotype. For a given animal, some haplotypes correspond to only one ERVK[2-1-LTR] element in its genome (which may be either C or D), while others may correspond to multiple ERVK[2-1-LTR] elements (which can be all C, all D or mixed C and D - such as A41 in the example). Different haplotypes may share the same variant for some tags (A63 and A33 share the same variant for tag 3 in the example). We typically only have partial PCIP-tag information for de novo insertions (f.i. only for the 5' or 3' LTR) potentially blurring the assignment of a de novo insertion to a specific haplotype. We therefore used an expectation-maximization (EM) algorithm to probabilistically estimate the proportional contribution of the different haplotypes to the de novo insertions.”

We have also added Suppl. Table 11 that provides the PCIP tag “haplotypes” for all ERVK[2-1-LTR] elements in all animals.

7. Figure 7b, c and d - please order the columns in the same way between the 3 plots, it is hard to compare between them in the current state.

Response 2.7

We are now showing the 23 haplotypes present in at least one of the three studied bulls in the same order for all three bulls, and have changed the figure legend accordingly.

“Figure 7: Preferential mobilization of D-type ERVK[2-1-LTR] elements. ... (b-d) ERVK[2-1-LTR] elements in the genome of three bulls were assigned to 13 (A), 17 (B) and 17 (C) haplotypes, for a total of 23 distinct haplotypes. The proportion of inherited ERVK[2-1-LTR] elements in each haplotype, as well as the proportion of C- (red) and D-type (blue) elements in each haplotype was unambiguously deduced by combining PCIP and targeted sequencing data, and is shown (23 haplotypes shown in the same order for the three bulls). The proportional contribution of each ERVK[2-1-LTR] haplotype to the de novo insertions was estimated by expectation-maximization (EM). As C- versus D-status cannot be directly

*deduced from PCIP-tag information, de novo insertions are represented in dark grey. The 95% confidence interval of the estimates (black lines) was determined by bootstrapping. (e-g) Proportion of C- and D-type elements amongst endogenous ("Const(itutive)", red) and de novo inserted ("Denovo", blue) ERVK[2-2-LTR] elements in the three bulls. De novo insertions mapped to mixed (C and D) haplotypes were distributed amongst C- and D-types according to the corresponding ratio of inherited elements. The black vertical bars in "Denovo" correspond to the 95% confidence interval determined by bootstrapping. The difference between the C/D ratio for inherited and de novo insertions was significant (** meaning $p < 0.01$ as determined by bootstrapping). "*

Minor points:

1. Please provide if possible as a supplementary figure a phylogeny (maybe as a circular plot) of all elements, based on multiple alignment. Metadata annotation around the plot would be insightful (allele frequency, category of recombinants, etc).

Response 2.8:

We have generated a phylogeny for all 309 ERVK[2-1-LTR] elements based on sequences (i) shared between the C and D clade, and (ii) devoid of evidence of recombination between the two clades. It is reported in Suppl. Fig. 2.

Supplementary Figure 2

a

b

- (a) **Dot plot between sequences representing the C clade (X-axis) and D clade (Y-axis).** The diagonal lines correspond to nearly identical 5' and 3'LTR, shared GAG and ENV segments (extremities), as well fragments of PRO and POL (*P* and *L* in main fig. 5). The limits of the GAG- (1) and ENV-shared (2) segments are boxed with dashed lines in the dot plot. The corresponding sequence alignments (1: GAG-shared; 2: ENV-shared) between a C and a D ERVK[2-1-LTR] element are displayed below the dot plot. Decreasing similarity (attrition) is marked by arrows with gradients (from high to low similarity).
- (b) **Neighbor-joining tree obtained with the concatenated GAG-shared (55 base pairs) and ENV-shared (195 base pairs) segments.** Indeed, these segments do not show obvious "within segment" traces of recombination between C and D-type elements so should provide the most correct relationship between C and D-type (see hereafter for one "between segment" recombination). Insertion-deletions of more than one nucleotide were collapsed to single events. C-type elements are highlighted in blue. C- and D-type elements appear as clear distinct clades. One C-element (chr. 24) is closer to the D-clade. It corresponds to the only element that has a complete C-type GAG-shared sequence associated with a D-type ENV-shared sequence. The average pairwise difference between C- and D-type elements was 38.4 in 250 base-pairs. Assuming a de novo mutation rate of 1×10^{-8} base pairs per generation [ref. 28], this would correspond to ~ 15 million generations or ~ 50 million years, hence suggesting that the endogenization of the exogenous retrovirus precursor of the ERVK[2-1-LTR] element is a very ancient event. It is, however, possible that some of the differences between C and D type shared sequences were introduced upon creation of D-type elements, and/or that the mutation rate for ERV elements, influenced by the retro-transposition process, is higher than 1×10^{-8} base pairs per generation. Estimates of the divergence are also influenced by the somewhat arbitrary definition of the boundaries of the GAG-shared and ENV-shared segments.

The corresponding information has been used in the **Discussion:** (lines 587-596) *"To gain additional insights in the history of the ERVK[2-1-LTR] clade we compared the shared GAG and ENV sequences between C- and D-type elements. We restricted the analysis to regions immediately flanking the boundaries of the C- and D-specific segments and presenting little or no evidence of recombination. The average pairwise difference between C- and D-type*

elements was 38.4 in 250 base-pairs. Assuming a de novo mutation rate of 1×10^{-8} base pairs per generation^[28], this would correspond to ~15 million generations or ~45 million years. It is, however, possible that some of the differences between C and D-type shared sequences were introduced upon creation of D-type elements, and/or that the mutation rate for ERV elements, influenced by the retro-transposition process, is higher than 1×10^{-8} base pairs per generation (Suppl. Fig. 2)."

We have determined the distribution of allelic frequencies for all C-type, D-type and solo-LTR elements in the 40 sequenced bulls. See also response 2.3

2. The tone of language through the manuscript when addressing transposable elements is overly negative. While I understand that the author's starting point is a new transposon causing disease, it is important not to skew readers perception to their positive potential. Examples:

"This mobilization of ERVs is a source of mostly deleterious variation"

"ERV endogenization has been caught in the act"

"We propose that D-type elements act as parasite-of-parasite"

Please update the text for a more balanced view of transposons, which should be more neutral (as most elements are) and not discount that they are important drivers of innovation.

Response 2.9:

We have modified the text in several parts of the manuscript to be more neutral with regards to the impact of ERVs on the organism's fitness (i.e. deleterious, neutral or advantageous). Specifically:

- (line 36) *"This mobilization of ERVs is a source of mostly deleterious variation" has been changed to "This mobilization of ERVs is a source of ~~mostly~~ deleterious variation".*
- (lines 74-79): *"Conversely, ERV elements provide a substrate for the emergence of new functionalities such as novel cis-acting regulatory elements^[3,6] and even new genes. As an example, the syncytin genes, which are essential for placentation, derive from ancient ERVs. Also, ERV elements may condition the host's susceptibility to exogenous retroviruses, including by providing protection. Hence, ERVs are important drivers of genomic innovation^[2-5]."*

We have elected to keep the "parasite-of-parasite" qualification as we think that it is an easy-to-grasp description of the phenomenon that we are trying to highlight.

3. Some precision in the language in some parts - for example:

"As an example, the syncytin genes, which are essential for placentation, are thought to derive from ancient ERVs"

Syncytin origin being an Env gene from an ancient ERV is well established, so "though to" is imprecise

2.10 Has been corrected (see also response 1.6).

4. Figure 2c undersells the size of the dataset - at first glance it looks like the whole pedigree is a few animals, and could mislead some readers. Please visually update, or add some small text on the figure panel with the number of animals at each level

2.11. Text has been added to the figure for clarification, as recommended.

5. "Five to ten percent of mammalian genomes is occupied by multiple families of endogenous retroviruses (ERVs), which may count thousands of dispersed members" - change to 'each of which may count'

2.12 Has been corrected.

6. "ERVs, leaving only "DNA fossils" attesting" - drop 'only'

2.13 Has been corrected.

7. "Halve of mammalian genomes" - "Half"

2.14 Has been corrected throughout manuscript (see also Response 1.5).

Reviewer #3 (Remarks to the Author):

Dear Authors,

First of all I have to apologize for my English. I am not practicing this language too often so do not expect smooth English or perfect use of words meanings or so. I try to do my best.

Before going to the review itself I have to admit that in my 35 year scientific life your paper was the most complicated, complex and demanding careful reading. Moreover, reading it was a pleasure because I felt page after page that I myself understand a new and fascinating phenomenon which I never was conscious.

Being captured by these feeling, I did not give up and tried to express my comments which I believe might be considered by Authors as having some value. They are written in spirit of sharing an independent view to what is the subject of the manuscript.

We thank the reviewer for these kind comments.

General remarks:

1. The title does not reflect exactly the results since the described phenomenon occurs primarily in Belgian Blue breed (although Holstein bulls were also used especially in population genetics analysis). So would suggest to extend the title, by adding at the end "...in Belgian Blue and Holstein bulls". I can admit that the problem is probably occurring in other breeds but the manuscript describes what happened in BB HOL breeds.

Response 3.1

We have studied the presence of ERVK[2-1-LTR] elements in other breeds and species. The results of this analyses are reported in the Discussion (lines 580 - 587): *"Sequences with similarities $\geq 95\%$ over $\geq 95\%$ of the full length of the chromosome 19 C-type element exist in the reference genomes of other domesticated taurine (*Bos taurus*) and indicine breeds (*Bos indicus*), gaur (*Bos gaurus*), gayal (*Bos frontalis*), domestic yak (*Bos grunniens*), wild yak (*Bos mutus*), bison (*Bison bison*), water buffalo (*Bubalus bubalis*), and African buffalo (*Syncerus caffer*). A marked drop in similarity is observed when querying the genomes of caprinae, including sheep (*Ovis aries*) and goat (*Capra hircus*) (Suppl. Table 15). This suggests that ERVK[2-1-LTR] endogenization may have occurred in an ancestor of Bovinae, i.e. ~ 15 million years ago."* (see also response 1.4). Hence, we think that mentioning cattle in the

title, is now supported by evidence. We have nevertheless slightly modified the **title** which now reads: *“GWAS reveals determinants of mobilization rate and dynamics of an active endogenous retrovirus of cattle.”* indicating that we do not study all bovine ERVs but only a specific, active “clade”.

2. I would appreciate very much to insert into MM section what was the genotype of Myostatin causing Muscle Hiperthrophy - do they have the same genotype or different, and if different what was the distribution of genotypes.

Response 3.2

We now specify the myostatin genotype of all used animals in **Methods**: (lines 908-912): *“Four hundred fifteen of the 430 utilized Belgian Blue animals were homozygous for the nt821(del11) mutation in the myostatin (MSTN) (gene causing double-muscling in homozygotes ^[42]), eight heterozygous, and seven homozygous wild-type. MSTN genotype had no effect on ERVK[2-1-LTR] mobilization rate (data not shown).”*

3. If I am not wrong, Belgian Blue cattle, and especially breeding bulls can be substantially related to each other (inbred). Therefore I would suggest to put inbreeding coefficient IC data into MM section. Moreover, if there are significant difference in IC within the bull population studied, I would suggest to take into account this variable into any statistical model used in this work to better recognize the factors influencing mobilisation rate and dynamics of endogenous retroviruses.

Response 3.3

We have tested the effect of inbreeding on the ERVK[2-1-LTR] mobilization rate. This is now reported in **Results** (lines 232-233): *“There was no evidence of an effect of the bulls’ inbreeding coefficient on ERVK[2-1-LTR] mobilization rate (Extended data Fig. 3c).”*, with accompanying Figure:

Extended data: Figure 3

Assessing the repeatability and robustness of PCIP.

(a) Correlation between the 5' and 3' PCIP estimates of the mobilization rate of ERVK[2-1-LTR] in sperm DNA of 430 Belgian Blue sires. The correlation of 0.68 indicates that PCIP is able to robustly measure differences in mobilization rate between samples. (b) Correlation between estimates of the mobilization rate of ERVK[2-1-LTR] elements in sperm samples for two biological replicates of ten young Belgian Blue bulls. Estimates correspond to the average of the 5'LTR and 3'LTR measures. Spearman's correlation was 0.85.

Lack of evidence of an effect of a bull's inbreeding coefficient on the rate of ERVK[2-1-LTR] mobilization in its germline.

(c) The ERVK[2-1-LTR] mobilization rate measured in sperm of 430 Belgian Blue bulls (Y-axis) as a function of the number of homozygous sites out of 7,428,183 SNPs (X-axis) ($MAF \geq 0.1$), used as proxy for their inbreeding coefficient.

And section in Methods (lines 927-930): *“Testing the effect of inbreeding on mobilization rate: The proportion of autosomal SNPs ($MAF \geq 0.1$) with homozygous genotype was used as proxy for a bull's coefficient of inbreeding. Its effect on mobilization rate was estimated by linear regression using the `lm()` R function.”*

4. From the same reason expressed in point 3, I would suggest to include into statistical model used throughout the work, the Estimated Breeding Value of bull (conventional or genomic) since this can also potentially be a factor somehow influencing mobilisation rate and dynamics of enogenous retroviruses.

Response 3.4

We agree with the reviewer. As a matter of fact, in addition to the effect of the tested SNPs/variants on mobilization rate, we are fitting a polygenic effect in the model to correct for stratification. This is mentioned in Results (lines 268-270): *“We conducted a GWAS for ERVK[2-1-LTR] mobilization rate using a linear model including a fixed variant dosage effect and a random polygenic effect to correct for stratification.”*

We specify in the **Methods** section how this was done exactly (lines 919-923): “We conducted GWAS with GEMMA^[43] using TR_S as phenotype. The model included a fixed regression on variant dosage (additive effect), as well as a random polygenic effect. Estimating the polygenic effects requires the additive genetic relationship matrix, which was computed from marker data using option “-gk1” in GEMMA.”

5. Almost all figures in main manuscript (and also all in Extended data) are very dense in information which is very positive, but any reader expects that all images tables or drawings must be visible enough for normal reading in printed A4 page. That is often impossible, especially Figure 3, 4 and 5. I would suggest to give one page for the the content of the figure, and transfer the Legend for next page.

Response 3.5

We have increased the resolution of all figures to the best of our ability and are now using as much space as possible for each in the submitted manuscript.

Specific comments:

Line 134 - explain shortly what is the difference between ERVK and ERVL

Response 3.6

We have added a paragraph in **Results** to clarify the composition of ERV elements in the bovine genome according to Repbase (lines 132-141): “Repbase^[14,15] reports four main groups of ERVs: ERVL-MaLR, ERVL, ERV1 and ERVK, jointly accounting for ~4.7% of genome space. While ERVL (ERVL-MaLR + ERVL) are the most abundant group in the reference genome (55.5% of genome space), followed by ERV1 (29.3%) and ERVK (15.2%), ERVK were the most abundant amongst polymorphic ERV elements (81.5% of polymorphic elements), followed by ERV1 (15.6%) and ERVL (2.9%). This is compatible (assuming approximately equal element size) with ERVK being younger and still active (Fig. 2a). Repbase reports 33 subgroups of ERVK, of which four are overrepresented amongst polymorphic ERVK elements, including ERVK[2-1-LTR] (20.2% of ERVK space, 26.6% of polymorphic ERVK elements) and BTLTR1B (10.2% of ERVK space, 27.1% of polymorphic ERVK elements) (Extended data Fig. 2a).”

The meaning of K in ERVK (and hence L in ERVL) is now addressed in the **Results** section: (lines 359 - 362): “The consensus primer binding site (PBS), shared by C-type and D-type elements (Suppl. Table 10), presents striking complementarities with the 3' extremities of the two bovine lysine tRNAs (CTT and TTT codons, respectively), supporting the ERVK denomination (Extended data Fig. 7c).”.

Line 136-137 - what statistical model generated this value?

Response 3.7

The p-values of the deviation between expected and observed proportions (genic vs intergenic and sense vs antisense) were determined by simulation in R. This is now explained in the legend to Fig. 2b (lines 172-173): “P-values were determined from the

outcome of $>10^6$ random samples with probabilities of success corresponding to the genomic expectations (sample function in R)."

The p-values for the shifts in DAF between gene vs intergenic and sense vs antisense insertions were determined using the Wilcoxon Sum of Rank test implemented with the ... function in R. This is now mentioned in the legend to Extended figure 2a&b.

Line 222 - SS shortcut should be explained here not later (line 228)

3.8. Has been corrected (line 240): "*... generating 5' and 3' shearing sites (SS) ...*".

Line 182 - explain in more detail understanding of "shared shearing sites"

3.9. We have added a sentence in the Results section to clarify the notion of "shared shearing sites" (lines 194-197): "*... (if two sperm cells carry the same de novo ERVK[2-1-LTR] insertion, the probability that the two corresponding DNA molecules are broken at the exact same position is assumed to be very low; shared SS are therefore assumed to correspond to PCR duplicates), ...*".

Line 211 - "more often" means statistically verified?

3.10. Indeed. This is why we provide p values between brackets. These were obtained using the same simulation approach described above.

Line 246 - put the method or software (and Reference) used for imputation of genotypes

3.11. The method used for imputation is described in detail in the methods section (lines 903 – 908): "*... genotype information was augmented by imputation in two steps (first to high density using 890 Belgian Blue animals genotyped with Illumina's high density (~770,000 variants) as reference, and then to whole genome using sequenced Belgian Blue animals as reference) using Shapit4^[40] for phasing, followed by Minimac4^[41] for actual imputation. We kept 10,875,490 variants with minor allele frequency over 2% and imputation accuracy over 90% for GWAS.*"

Line 253 - I would like to see the SNP cluster image, its quality (add to Supplementary section)

3.12. The lead SNPs were all imputed SNPs for which we have no cluster images. However, we have added to the legends of Fig. 4 and extended Fig. 5 that "*All non-ERVK lead variants were imputed variants with imputation accuracy ≥ 0.95* "

Line 252 - I again suggest to enrich the linear model used in GWAS with for me obvious variables, eg. relatedness matrix, sire effect, EBV

3.13. See response 3.4. This is a state-of-the-art model.

Line 591 - put the Reference - who is the author of "parasite-of-parasite" concept and explain in few sentences this concept.

3.14. We assume that other people have also used the "parasite-of-parasite" quote but could not find a princeps reference that may have introduced this term.

REVIEWERS' COMMENTS

Reviewer #1 (Remarks to the Author):

This already experimentally strong manuscript has, in my opinion, been greatly improved by the authors' reinterpretation of their results away from the highly improbable "retrotransposition" model in favor of the far more reasonable reinfection model, for which they now provide evidence. There are, however, still a couple of points of interpretation where they don't fully appreciate the implications of this mechanism of "transposition."

First, the mechanism of retroviral reverse transcription. RNase H degradation of the RNA template shortly after it is copied allows the growing DNA strand to transfer to another template when synthesis reaches the template end. This step must occur twice to form the LTR, but it also happens if the enzyme encounters an internal break in the genome, allowing the virus to recover from what would otherwise be lethal damage. Such breaks are quite common, accounting for very high recombination frequencies observed in cell culture experiments. It should be explicitly presented in the discussion of recombination, Lines 669ff. It also provides a plausible mechanism for the apparent dominance of acquisition D over C-type proviruses during mobilization, since, during reverse transcription of a heterozygous (C/D) dimer, a break in the D genome could always be recombinationally repaired, but one in the unique 40% of the C genome could not and would be lethal to the virus. This possibility should be discussed around lines 672-676.

Second, another important property of reverse transcription is its error rate, which is on the order of a few times 10⁻⁵ mutations per copying, or more than 1000-fold higher than the cellular DNA replication error rate. This consideration sets a minimum distance between parental and daughter proviruses. Added to this is the probability of multiple replication cycles associated with somatic cell-cell spread between the initial appearance of the virus and the infection of the germline cell, further compounding the age estimate. For this reason, distance between non-allelic proviruses will always overestimate – quite likely by a large factor – the time since integration. Thus, the LTR-LTR distance is, by far, the more reliable indicator of proviral age. The discussion on the first paragraph of page 23 should be rewritten to acknowledge this. Also, the sentence on lines 619-621 makes no sense. Once a solo LTR is formed, there is no way for it to revert to a full-length 2-LTR provirus, so it would be useless for age calculation.

Given the above, it would be of great interest to test the likelihood of somatic spread of the progeny of one of the intact proviruses by using the PCIP assay on cells other than sperm, such as in blood and non-germline reproductive tissue. It would also be of interest to express the intact C type provirus in cell culture. To study its replicative properties. Have the authors done so?

Third, the mention of piRNAs on lines 687-690 also misses the mark for reinfection-mediated mobilization, since the germline-expressed suppressive RNAs would be expected to have no effect on the reverse transcription and integration steps, which would be the only ones that occur in germline cells following infection with virus produced by a somatic cell.

Other points:

Line 46: "non-solo" is non-standard. Usually referred to as "full length" or "2-LTR" proviruses.

Line 60: Actually, retrovirus budding is independent of Env.

Line 61: "subsequently" and rarely.

Lines 133-143: This discussion treats these groups as if they were monolithic. In fact, each, especially ERVK, which is defined by the (taxonomically nearly useless) imputed tRNA primer, comprises many subgroups, of little relationship to one another and dissimilar in age and other properties. It would be more accurate on lines 135-137 to say that ERVK includes the most abundant polymorphic elements.

Line 362-363: Not all retroviruses have myristylated Gag. Alpharetroviruses do not and still assemble and bud from the plasma membrane.

Line 366 and Figure S7c: "striking complementarities" Given the mechanism by which primer binding sites are copied, it is quite odd that they are not perfectly identical to the complement of the lys tRNA.

Line 548-550 and 695-697: Another mechanism of extinction, known to have occurred in many species, including humans, is the appearance of an ERV that expresses a normal or mutant gene product (especially Env) which blocks replication of the cognate virus (See for example Blanco-Melo D, 2017. Co-option of an endogenous retrovirus envelope for host defense in hominid

ancestors. Elife 6).

Figure S6: This is useful but needs some fixing. First, the Gag amino acid sequence has been cut off and is missing about 4 lines. Second, Please use standard retroviral nomenclature for the genes, domains, and motifs:

Gag: MA, CA (NTD, CTD), NC.

Pro: DU, PR

Pol: RT, IN

Env: SU, TM

It would also be helpful to indicate more motifs, such as the late domain (eg PTAP or others) probably in the white portion of Gag, the major homology region in CA, and the YMDD active site of RT. italics

Finally, what is the meaning of the italics at the beginning of the PR amino acid sequence?

Figure S7a: It would be interesting to also include the most recent human ERV, HERV-K (HML-2), LTR5Hs in the trees.

Minor issues:

Line 38: "of" should be "to."

Line 42: "between" should be "among"

Line 676: Define "ecDNA"

Reviewer #2 (Remarks to the Author):

All my previous comments were addressed, I am now happy to recommend the manuscript for publication.

RESPONSE TO REVIEWERS' COMMENTS

Reviewer #1 (Remarks to the Author):

This already experimentally strong manuscript has, in my opinion, been greatly improved by the authors' reinterpretation of their results away from the highly improbable "retrotransposition" model in favor of the far more reasonable reinfection model, for which they now provide evidence.

There are, however, still a couple of points of interpretation where they don't fully appreciate the implications of this mechanism of "transposition."

First, the mechanism of retroviral reverse transcription. RNase H degradation of the RNA template shortly after it is copied allows the growing DNA strand to transfer to another template when synthesis reaches the template end. This step must occur twice to form the LTR, but it also happens if the enzyme encounters an internal break in the genome, allowing the virus to recover from what would otherwise be lethal damage. Such breaks are quite common, accounting for very high recombination frequencies observed in cell culture experiments. It should be explicitly presented in the discussion of recombination, Lines 669ff. It also provides a plausible mechanism for the apparent dominance of acquisition D over C-type proviruses during mobilization, since, during reverse transcription of a heterozygous (C/D) dimer, a break in the D genome could always be recombinationally repaired, but one in the unique 40% of the C genome could not and would be lethal to the virus. This possibility should be discussed around lines 672-676.

We thank the reviewer for this interesting comment. However, we don't fully understand the basis for the suggestion that a break in the D-type gRNA would be more prone to repair by recombination than a break in the C-type gRNA, in heterozygous C/D particles. At best, the D-type gRNA would be less likely to harbor a break than its C-type counterpart (because of its smaller size), or a break would be less likely to affect the D-type specific central segment than the C-type specific central fragment because of its smaller size and/or smaller proportion of the cognate gRNA. Also, our best estimates of the "recombination rate" is of the order of ~ 5 in 400 (see lines 351-353), which seems too low to make a significant contribution to the observed over-representation of D-type elements amongst de novo mobilized elements. Nevertheless, the idea that heterozygous C/D particles might produce more D- than C-type cDNA is intriguing and we have added that idea to the discussion (lines 497-499): *"Another intriguing possibility would be that "heterozygous" virus-like particles harboring a C- and a D-type gRNA would preferentially produce D-type double stranded extracellular DNA."*

Second, another important property of reverse transcription is its error rate, which is on the order of a few times 10^{-5} mutations per copying, or more than 1000-fold higher than the cellular DNA replication error rate. This consideration sets a minimum distance between parental and daughter proviruses. Added to this is the probability of multiple replication cycles associated with somatic cell-cell spread between the initial appearance of the virus and the infection of the germline cell, further compounding the age estimate. For this reason, distance between non-allelic proviruses will always overestimate – quite likely by a large factor – the time since integration.

The reviewer is right and this concern is now addressed in the Discussion, lines 427-430: *"However, this figure doesn't account for the extra mutations introduced during the undetermined number of reputedly low-fidelity reverse transcription steps that separate the different C- and D-type elements considered in this analysis. It is therefore bound to be overestimated, possibly grossly."*

Thus, the LTR-LTR distance is, by far, the more reliable indicator of proviral age. The discussion on the first paragraph of page 23 should be rewritten to acknowledge this. Also, the sentence on lines 619-621 makes no sense. Once a solo LTR is formed, there is no way for it to revert to a full-length 2-LTR provirus, so it would be useless for age calculation.

We have revisited the analysis of the similarity between the 5' and 3' LTR as a way to estimate the age of the ERVK[2-1-LTR] clade, and are commenting on it in the Discussion (lines 430-443): *"We also analyzed the sequence divergence between the 5' and 3' LTRs of all C and D-type elements. Upon creation of a new insertion, 5' and 3'LTR have identical sequence. Observed differences therefore reflect the accumulation of de novo mutations and therefore presumably provide information about the time of insertion. The full-length ERVK[2-1-LTR] element with the most divergent 5' and 3'LTRs (98.3% similarity over 1,287 base pairs) is a C-type element on chromosome 6, present in the bovine reference genome and fixed in the Belgian Blue cattle population (Suppl. Table 8). Assuming a mutation rate of 1×10^{-8} base pairs per generation, this level of divergence is expected to accrue over ~850,000 generations or ~2.5 million years, hence supporting a more recent time of primordial endogenization of the ERVK[2-1-LTR] clade than the two previous estimates. It should be noted, however, that the eldest ERVK[2-1-LTR] elements, whose 5' and 3'LTR comparison would best inform about the time of origin of the clade, are more likely to have been reduced to solo-LTRs. This third figure is therefore liable to be an underestimate."*

Given the above, it would be of great interest to test the likelihood of somatic spread of the progeny of one of the intact proviruses by using the PCIP assay on cells other than sperm, such as in blood and non-germline reproductive tissue. It would also be of interest to express the intact C type provirus in cell culture. To study its replicative properties. Have the authors done so?

We have (interesting) preliminary results on somatic non-reproductive and reproductive tissue. Yet we consider that this should be the topic of a future, distinct manuscript. We have not yet studied the C type provirus in culture.

Third, the mention of piRNAs on lines 687-690 also misses the mark for reinfection-mediated mobilization, since the germline-expressed suppressive RNAs would be expected to have no effect on the reverse transcription and integration steps, which would be the only ones that occur in germline cells following infection with virus produced by a somatic cell.

We agree with the reviewer and have eliminated the reference to piRNAs from this section of the manuscript.

Other points:

Line 46: "non-solo" is non-standard. Usually referred to as "full length" or "2-LTR" proviruses.

Has been corrected.

Line 60: Actually, retrovirus budding is independent of Env.

We have rephrased this as (lines 55-56): *"ERV-encoded envelope (ENV)-associated budding of viral particles followed by germline reinfection."*

Line 61: "subsequently" and rarely.

We have added "sometimes"

Lines 133-143: This discussion treats these groups as if they were monolithic. In fact, each, especially ERVK, which is defined by the (taxonomically nearly useless) imputed tRNA primer, comprises many subgroups, of little relationship to one another and dissimilar in age and other properties. It would be more accurate on lines 135-137 to say that ERVK includes the most abundant polymorphic elements.

Has been corrected accordingly (“includes”)

Line 362-363: Not all retroviruses have myristylated Gag. Alpharetroviruses do not and still assemble and bud from the plasma membrane.

We have removed “essential to address viral particles of both exogenous and endogenous retroviruses to the cell membrane;” from the Discussion.

Line 366 and Figure S7c: “striking complementarities” Given the mechanism by which primer binding sites are copied, it is quite odd that they are not perfectly identical to the complement of the lys tRNA.

Mismatches between primer binding sites and cognate tRNA 3' ends have been reported before and are thought to reflect “the fine line between priming and silencing walked by endogenous retroviruses” (Cullen & Schorn, 2020). This is now added to the results section (lines 290-292) with accompanying reference: “*Observed mismatches might reflect the required balance between adequate tRNA priming for effective reverse transcription yet reduced complementarity to tRNA fragments mediating silencing [27].*”.

Line 548-550 and 695-697: Another mechanism of extinction, known to have occurred in many species, including humans, is the appearance of an ERV that expresses a normal or mutant gene product (especially Env) which blocks replication of the cognate virus (See for example Blanco-Melo D, 2017. Co-option of an endogenous retrovirus envelope for host defense in hominid ancestors. Elife 6).

The corresponding mechanism is now mentioned using the Blanco-Melo reference in the introduction (line 67): “... and so-called restriction factors targeting various steps of the ERV life cycle, some of which are co-opted ERV genes ... [2-6]”.

Figure S6: This is useful but needs some fixing. First, the Gag amino acid sequence has been cut off and is missing about 4 lines.

Has been corrected

Second, Please use standard retroviral nomenclature for the genes, domains, and motifs:

Gag: MA, CA (NTD, CTD), NC.

Pro: DU, PR

Pol: RT, IN

Env: SU, TM

It would also be helpful to indicate more motifs, such as the late domain (eg PTAP or others) probably in the white portion of Gag, the major homology region in CA, and the YMDD active site of RT.

We now have added the standard nomenclature to the Supplementary Figure 9 whenever possible and we have highlighted the YMDD active site of RT.

We also have annexed the Supplementary Table 1 describing domain annotations for each of the four proteins (GAG, PRO, POL and ENV).

Finally, what is the meaning of the italics at the beginning of the PR amino acid sequence?

Amino acids in italics at the N-terminal part of the PRO protein correspond to putative translation start sites. This is now explained in the legend.

Figure S7a: It would be interesting to also include the most recent human ERV, HERV-K (HML-2), LTR5Hs in the trees.

We have performed the analysis requested by the reviewer which – as shown hereafter - indicates that HML2 clusters with the beta-retroviridae. We have elected not to replace the original figure as the message doesn't change.

Minor issues:

Line 38: "of" should be "to."

Has been done

Line 42: "between" should be "among"

Has been done

Line 676: Define "ecDNA"

Has been defined (line 493)